



# Long-term water clarity patterns of lakes across China using Landsat
series imagery from 1985 to 2020
Xidong Chen [1], Liangyun Liu [2, *], Xiao Zhang [2, 3], Junsheng Li [2, 3], Shenglei Wang [2], Yuan Gao
[4], Jun Mi [2, 3]

5         1. North China University of Water Resources and Electric Power, Zhengzhou 450046, China;

2. Key Laboratory of Digital Earth Science, Aerospace Information Research Institute, Chinese Academy

7                              of Sciences, Beijing 100094, China;

8                   3. University of Chinese Academy of Sciences, Beijing 100049, China;

9                        4. Beijing Normal University, Beijing 100091, China

10             ∗ Corresponding author: Liangyun Liu, E-mail address: liuly@radi.ac.cn.

**Abstract:** Monitoring the water clarity of lakes is essential for the sustainable development of
human society. However, existing water clarity assessments in China have mostly focused on
lakes with areas > 1 km², and the monitoring periods were mainly in the 21st century. In order
to improve the understanding of spatiotemporal variations in lake clarity across China, based
on the Google Earth Engine cloud platform, a 30 m long-term LAke Water Secchi Depth (SD)
dataset (LAWSD30) of China (1985–2020) was first developed using Landsat series imagery and
a robust water-color-parameter-based SD model. The LAWSD30 dataset exhibited a good
performance compared with concurrent in situ SD datasets, with an $R^2$ of 0.86 and a root-mean-
square error of 0.225 m. Then, based on our LAWSD30 dataset, long-term spatiotemporal
variations in SD for lakes > 0.01 km² ($N$ = 40,973) across China were evaluated. The results show
that the SD of lakes with areas ≤ 1 km² exhibited a significant downward trend in the period
1985–2020, but the decline rate began to slow down and stabilized after 2001. In addition, the
SD of lakes with an area > 1 km² showed a significant downward trend before 2001, and began
to increase significantly afterwards. Moreover, in terms of the spatial patterns, the proportion
of small lakes (area ≤ 1 km²) showing a decreasing SD trend was the largest in the Mongolian–
Xinjiang Plateau Region (MXR) (about 30.0%), and the smallest in the Eastern Plain Region (EPR)
(2.6%). In contrast, for lakes > 1 km², this proportion was the highest in MXR (about 23.0%), and
the lowest in the Northeast Mountain Plain Region (NER) (16.1%). The LAWSD30 dataset and
the spatiotemporal patterns of lake water clarity in our research can provide effective guidance
for the protection and management of lake environment in China.
**Keywords:** water clarity; Secchi Depth (SD); Landsat; Google Earth Engine; long-term
**1. Introduction**

34         Lakes are invaluable resources for human societies, providing value in terms of water

supply, energy production, commerce, food production, and human health (Bastviken et al.,
2011; Palmer et al., 2015). However, like many other ecosystems, lakes are sensitive to multiple
co-occurring environmental pressures, notably climate change, nutrient enrichment, organic
and inorganic pollution, and human activities (Brönmark and Hansson, 2002). Nowadays, with
the rapid development of the economy and the growth of the population in China, the
intensification of human activities and pollution from industry and agricultural production



have caused severe damage to lakes (Ma et al., 2014; Wang and Yang, 2019; Zhou et al., 2019).
According to recent research and national survey reports (Barnes, 2014; Wang and Yang, 2019;
Ministry of Ecology and Environment of the People's Republic of China, 2020), approximately
70% of inland water in China is polluted, 28% of the assessed lakes are eutrophic, and about 140
million people depend on getting water from unsafe open sources. The deterioration of the lake
ecosystem has threatened public health and the safety of both humans and aquatic organisms
(Guo, 2007). Therefore, effective monitoring and evaluation of the environment of lakes across
China is necessary.
Water clarity is one of the most intuitive, popular, and important parameters for describing
the optical components of water bodies (Carlson, 1977; Liu et al., 2020), and is generally
measured in terms of Secchi Depth (SD) (Odermatt et al., 2012; Carlson, 1977). Since water clarity
is co-determined by the suspended matter, planktonic algae, and colored dissolved organic
matter in the water column, it is usually adopted as a practical comprehensive metric for water
quality assessment (Kloiber et al., 2002; Mccullough et al., 2012). Although a variety of physical,
biological, and chemical parameters have been proposed to analyze the condition of water,
water clarity has been utilized for a century as an effective and simple metric (Cuffney et al.,
2000; Lee et al., 2018). Recently, water clarity was also recognized as an important parameter in
support of the United Nations Sustainable Development Goal SDG 6.3.2 evaluation reports
(Shen et al., 2020). Therefore, water clarity is a significant indicator that can be used to monitor
and evaluate the comprehensive conditions of water.
Today, with the development of remote sensing technology, significant numbers of satellite
images are continuously being acquired. Taking into account the high-frequency revisits, large
area of coverage, and the historical archive of remote sensing data available, increasing attention
has been paid to the applications of remote sensing datasets in water clarity assessments (Li et
al., 2020a; Liu et al., 2019; Xue et al., 2019). The evaluation of water clarity from a variety of ocean
color satellite sensors has been performed (Li et al., 2020a; Feng et al., 2019; Wang et al., 2018;
Shi et al., 2018). For example, Shen et al. (2020) used Sentinel-3 data to evaluate the water clarity
of 86 lakes (> 30 km$^2$) in eastern China; Liu et al. (2021) estimated the SD (water clarity) trends
of lakes with an area > 50 km$^2$ in the Tibetan Plateau using MODIS data between 2000 and 2019.
However, due to the coarse spatial resolution and the relatively short-term historical archives of
these ocean color sensors, their applications were limited to large lakes and reservoirs, and the
study periods were concentrated in the past two decades (Li et al., 2020a). The statistics on lakes
with an area ≤ 1 km$^2$ are scant, and understanding of the variations in SD before the 21st century
is limited (Downing et al., 2012; Biggs et al., 2017; Li et al., 2020b). In order to improve the water
environment monitoring capability, 30 m Landsat series data have recently been used for SD
evaluation (Page et al., 2019; Dona et al., 2014). Because of the fine spatial resolution (30 m), long
historical archives (> 35 years), and suitability for water clarity assessment, Landsat series data
are considered to be "ideal" for the long-term and fine spatial resolution monitoring of lake SD
(Olmanson et al., 2008; Olmanson et al., 2016; Li et al., 2020a; Zhang et al., 2021b). For example,
Li et al. (2020a) utilized the Landsat series of images to monitor the SD trends in the Xin'anjiang
Reservoir between 1986 and 2016; Yin et al. (2021) tracked the SD changes in Taihu from 1984 to
2018 based on Landsat 5 and 8 images. Recently, in order to conduct the first high-spatial-
resolution investigation of lake SD across China, significant amounts of Landsat 8 data from
2014–2017 were used in the work of Song et al. (2020). However, these studies were limited to





individual areas or periods. Due to the requirement for huge amounts of computation and large
storage capabilities, as well as the need for a robust uniform SD model, there are very few
examples of national-scale long-term SD estimations using Landsat imagery (Yin et al., 2021;
Kloiber et al., 2002; Page et al., 2019).
Fortunately, with the emergence of the Google Earth Engine (GEE) cloud computing
platform (Gorelick et al., 2017), its high-performance, intrinsically parallel computing services
can easily meet the requirements for very large computational resources (Zhang et al., 2020;
Liangyun Liu et al., 2021). Additonally, because the GEE platform integrates multipetabyte
analysis-ready Landsat surface reflectance data, and these data are intercalibrated between
different Landsat sensors, it presents an opportunity to conduct long-term land surface analyses
at the pixel level (Racetin et al., 2020; Zhang et al., 2021b). Accordingly, a robust SD model is the
only requirement for fine-resolution, long-term SD evaluation across China. Lately, some
studies have found that the SD is well correlated with water color parameters (e.g., hue angle
and the Forel–Ule Index (FUI)) (Wang et al., 2021; Chen et al., 2021; Van Der Woerd and Wernand,
2018). Since water color parameters can be retrieved at the global scale and over long time spans
(Wang et al., 2021; Wang et al., 2018), it is possible to retrieve long-term water clarity over large
areas based on these parameters (Wang et al., 2021; Wang et al., 2020). For example, Wang et al.
(2020) recently developed a robust SD model based on water color parameters, and the model
was successfully applied to MODIS data to develop a nationwide 500 m long-term SD dataset
between 2000 and 2017. Accordingly, a feasible solution for high-spatial-resolution and long-
term SD estimation across China could be provided by incorporating the GEE cloud platform
and the water-color-parameter-based SD model.
Therefore, in order to provide a comprehensive understanding of nationwide
spatiotemporal variations in lake water clarity, we first developed a long-term 30 m LAke Water
SD dataset of China from 1985 to 2020 (LAWSD30) using Landsat series data and a water-color-
parameter-based SD algorithm with the assistance of the GEE cloud platform. Then, the
LAWSD30 dataset was employed to evaluate and recognize the spatiotemporal variations in SD
for lakes with areas > 0.01 km² ($N$ = 40,973) across China in the period 1985–2020. Our results
can provide effective data support for the management and protection of lake water
environment.

## 2. Datasets

### 2.1. Landsat series satellite datasets

Taking into account the frequent contamination of cloud and cloud shadow, it is hard to
develop a spatially continuous product throughout China with only one year of Landsat images
(Zhang et al., 2019a). Therefore, we used images from ± 1 year of the target year to generate each
product, and a total of 12 SD products with a three-year time step were developed for 1985–
2020. All available Landsat series Level-1 precision terrain (L1TP) surface reflectance datasets
(about 46 terabytes of data), including Landsat 5 Thematic Mapper (TM), Landsat 7 Enhanced
Thematic Mapper-plus (ETM+), and Landsat 8 Operational Land Imager (OLI) imagery,
acquired in the summer (June 1–September 30) from 1985 to 2020, were used via the GEE cloud
computing platform. The summer months were chosen because the water clarity is relatively
stable in this season and suitable for monitoring with remote sensing imagery (Kloiber et al.,
2002; Mccullough et al., 2012; Singh and Singh, 2015; Song et al., 2020). In addition, since the





Landsat L1TP data were intercalibrated across different Landsat series sensors, the collected
L1TP data were consistent and suitable for pixel-level time series analysis (Racetin et al., 2020).
However, since the Landsat 7 scan line corrector (SLC) failed in 2003, the Landsat 7 images
acquired thereafter exhibited wedge-shaped scan gaps (referred to as SLC-off images) (Usgs,
2003). Therefore, except for 2012–2014, only Landsat 7 data before 2003 were used for the
development of our SD products (Fig. 1b). Since Landsat 5 retired in 2011 and Landsat 8 data
were only available after 2013, the valid Landsat observations from 2012 to 2014 were
insufficient (Fig. 1a). Therefore, a few Landsat 7 SLC-off images from 2012 to 2014 were used as
substitutes to fill the gaps between 2012 and 2014. Fig. 1b shows the final number of Landsat
series images used to generate the SD product for each nominal year.

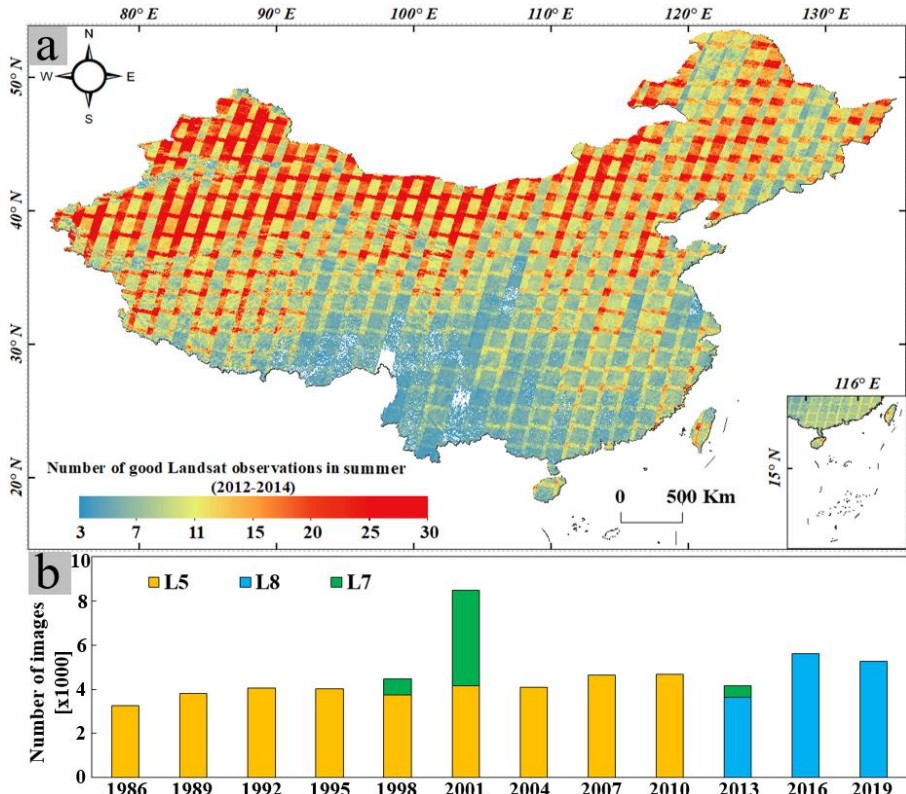


**Figure 1.** Valid Landsat 5 and 8 observations in China from 2012 to 2014 (**a**) and statistics of Landsat images
used to develop the product for each nominal year (**b**). Note: L8., Landsat 8; L7., Landsat 7; L5., Landsat 5.
*2.2. Auxiliary inland water products*

142        The annual 30 m Joint Research Centre Global Surface Water (JRC-GSW) database was used
to extract water body regions for each SD product (Pekel et al., 2016). The JRC-GSW was
developed based on multiple classification criteria and time-series Landsat 5, 7, and 8 data from
1984 to 2019, and archived to the GEE platform. The water pixels in the JRC-GSW were labeled
as permanent and seasonal water pixels based on the frequency of being detected as water
bodies (Pekel et al., 2016). The overall user and producer accuracies for permanent water were
99.6% and 98.6%, respectively, versus 98.6% and 75.4% for seasonal water (Pekel et al., 2016).





Following Chen et al. (2021), in this study, only the pixels marked as permanent waters in the
JRC-GSW were utilized to extract water regions to reduce the disturbance from aquatic
vegetation in seasonal waters.
Additionally, the existing Chinese lake inventories (Ma et al., 2011; Song et al., 2020; Chen
et al., 2021) and the Reservoirs and Dams vector database (Song et al., 2018) were also collected
and used to extract lakes for each SD product.
*2.3. In situ SD datasets*
In order to quantitatively evaluate the performance of the LAWSD30 dataset, a total of 1502
in situ SD measurements of 208 lakes between 1992 and 2019 were collected from the China Lake
Scientific Database (http://www.lakesci.csdb.cn), the National Earth System Science Data Center,
National Science & Technology Infrastructure of China (http://lake.geodata.cn), and work by
Wang et al. (2020) and Liu et al. (2020). Due to the scarcity of field-measured SD records before
the 1990s, only SD products after 1992 were validated. Since in situ SD measurements within
seven days of satellite overpasses were suitable for the validation of the remote-sensing-derived
SD product (Song et al., 2020), the collected SD measurements were coincident with the Landsat
data used in our study within a window of ± 7 days. The distributions of the in situ SD records
collected to validate products for different nominal years are shown in Fig. 2a–c. The probability
density of the collected SD measurements used for each SD product was calculated and is
exhibited in Fig. 2d. It can be seen that the collected SD measurements cover a variety of water
clarity conditions and are distributed throughout China (Fig. 2). The values of our collected SD
data range from 0 to 7 m, covering lakes from clear to eutrophic. Therefore, the collected in situ
data can provide a reliable accuracy examination for our LAWSD30 dataset.

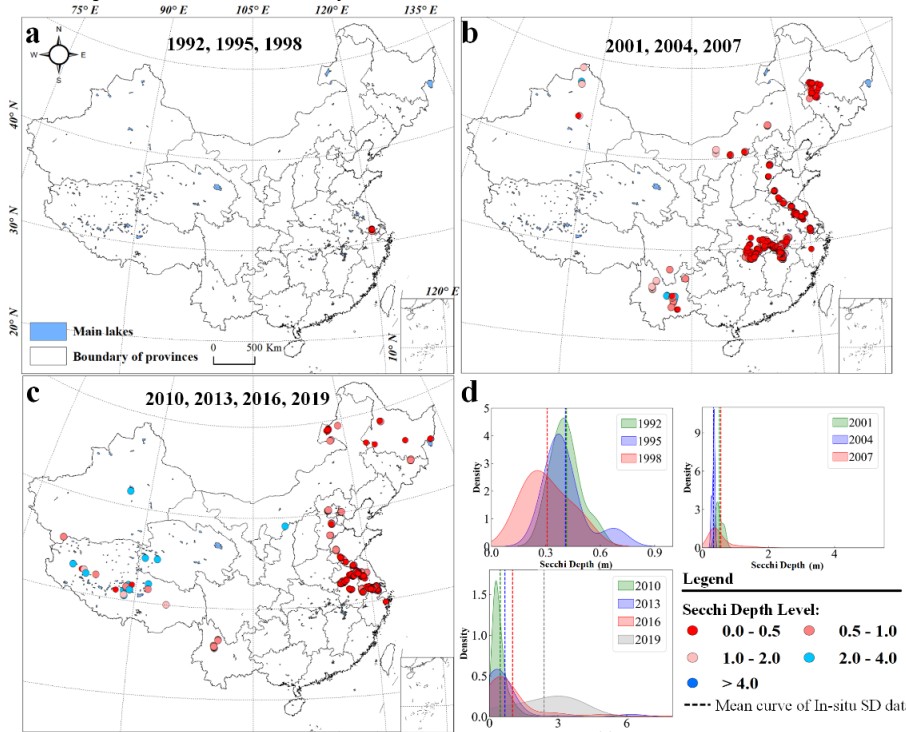






**Figure 2.** Details of the in situ measured SD datasets. (**a–c**) The geographical distributions of SD samples
used to validate the accuracies of the corresponding SD products; (**d**) the probability density of the
collected SD measurements used for each target SD product, used to show the SD range where the collected
in situ SDs are mainly concentrated.
**3. Methodology**
In order to assess the long-term trends of SD in Chinese lakes, four steps were taken in our
study (Fig. 3). First, based on the time-series Landsat images and the JRC-GSW water products
archived in GEE, a summer cloud-free composite image was generated between 1985 and 2020
with an interval of three years using the best-available-pixel (BAP) compositing method. Then,
based on the generated cloud-free composite images, the long-term LAWSD30 dataset from
1985 to 2020 (including 12 products, representing 1986, 1989, 1992, 1995, 1998, 2001, 2004, 2007,
2010, 2013, 2016, and 2019) were developed using a robust SD model based on water color
parameters. Next, the accuracy of our developed LAWSD30 dataset was evaluated using the
collected concurrent in situ datasets. Finally, with the assistance of the existing Chinese lake and
reservoir datasets and high-resolution images in Google Earth, the assessment of the long-term
SD trends for lakes > 0.01 km² across China was conducted using the developed SD products.

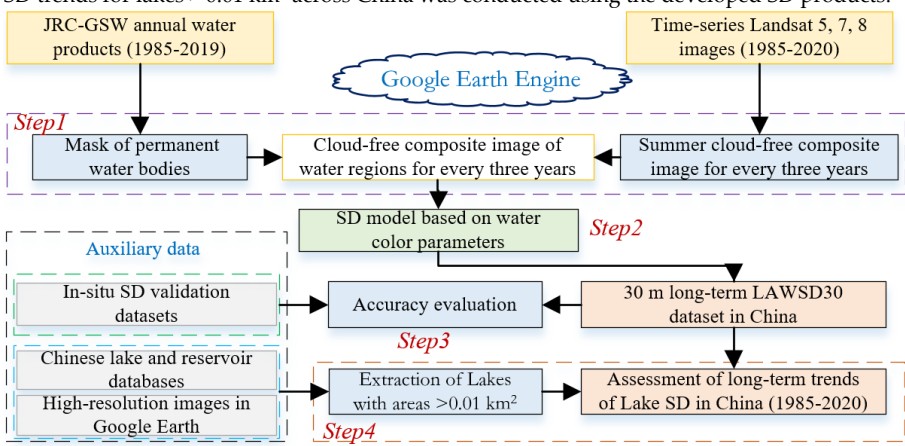


**Figure 3.** A flowchart of the long-term LAWSD30 dataset development steps and the long-term SD trend
assessment of lakes in China. Note: LAWSD30., 30 m LAke Water SD dataset of China.
*3.1. Generation of cloud-free composites using best-available-pixel (BAP) composition method*
Since summer is suitable for SD mapping with remote sensing imagery (Mccullough et al.,
2012; Singh and Singh, 2015; Song et al., 2020), cloud-free summer composites of Landsat data
for 12 three-year time steps were compiled from 1985 to 2010. Generally, the median and mean
composite methods were used to generate cloud-free images in the SD assessing studies (Li et
al., 2020a; Wang et al., 2020; Liu et al., 2020). However, since multisource sensors (TM, ETM+,
and OLI) were used in this study, the different band settings between these sensors made the
parameters of the algorithm specific to each sensor (Li et al., 2020a; Garnesson et al., 2019).
Because the median and mean composite methods will change the original radiation value of
the pixel (Xie et al., 2019; Griffiths et al., 2014), the composite pixels derived from these general
methods are difficult to trace back to the sensor from which they originated. Therefore, these
methods are not well suitable for our research. Recently, White et al. (2014) proposed a BAP





method to generate cloudless composites on a large area. Since BAP compiles cloud-free images
by selecting the best available observation based on user-defined criteria (Gomez et al., 2016;
Griffiths et al., 2013), the BAP composites can retain the source image information from which
they came. In addition, since BAP can ensure phenological consistency between multitemporal
BAP composites by setting the acquisition day-of-year (DOY) criteria (Griffiths et al., 2014; Chen
et al., 2021), it is suitable for multiyear change detection and assessment (Griffiths et al., 2014;
Gomez et al., 2016; Hermosilla et al., 2015; Zhang et al., 2021a). Accordingly, the BAP method
was used to generate cloud-free composites. Our team recently used BAP to develop summer
composites for water color mapping (Chen et al., 2021). Following him, the DOY criteria, the
cloud and cloud shadow criteria, and the atmospheric opacity criteria were selected to generate
the BAP composites. The score for each criterion was summed, and the observation with the
highest score was selected as the BAP composite. The parameter values for the criteria used were
obtained from Chen et al. (2021).
However, since floods and rainfall in summer will bring suspended particles into water
bodies, making the SD of water bodies much lower than usual (Murshed et al., 2014; Liu et al.,
2021), it is also necessary to reduce the impact of these factors on the BAP composites to ensure
the reliability of the long-term SD trend assessment. Here, the normalized difference turbidity
index (NDTI) (Lacaux et al., 2007) was used to indicate the turbidity of the water (Eq. (1)). As
the SD of water decreases, the NDTI of water increases (Islam, 2006; Lacaux et al., 2007).
Therefore, the interference of floods and rainfall was restricted by only using the observations
with NDTI less than the 80th quantile of their NDTI stack for BAP compositing.

$$NDTI = (Red - Green)/(Red + Green) \tag{1}$$

Finally, based on the intra-annual permanent water pixels detected in the JRC-GSW, water
regions were extracted from the BAP composites.
*3.2. Inversion model of water SD*
Previous studies have proven that FUI and hue angle ($\alpha$) are useful water color parameters
for assessing the SD of inland waters (Wang et al., 2020; Chen et al., 2021; Garaba et al., 2015).
Recently, these two watercolor parameters were further demonstrated to be robust parameters
for retrieving SD over large areas and long-term spans (Wang et al., 2020; Pitarch et al., 2019).
Therefore, the SD of the extracted permanent water regions was retrieved using a robust SD
model based on FUI and $\alpha$ (Wang et al., 2020). The SD model showed good performance and
adaptability over a variety of water clarity ranges, with a mean relative difference of 27.4% and
a mean absolute difference of 0.37 m (Wang et al., 2020). There are three main steps in the SD
model:
(1) *Calculation of the hue angle ($\alpha$)*: The $\alpha$ is the angle of the line drawn anti-clockwise from
the positive x-axis at y = 1/3 in the Commission on Illumination's (CIE) chromaticity diagram
(Wang et al., 2018). In order to derive the angle $\alpha$, the CIE primary color tristimulus (X, Y, Z)
was calculated from the reflectance in the visible bands of Landsat images first (Wang et al., 2020;
Chen et al., 2021). Since ETM+/TM has only three bands in the visible range, the tristimulus of
Landsat ETM+/TM was calculated using the RGB conversion method (Wang, 2018; Cie, 1932)
(Eqs. (2)-(4)):

$$X = 1.1302 \ R(485) + 1.7517 \ R(565) + 2.7689 \ R(660) \tag{2}$$

$$Y = 0.0601 \ R(485) + 4.5907 \ R(565) + 1.0000 \ R(660) \tag{3}$$



$$Z = 5.5943 \ R(485) + 0.0560 \ R(565), \tag{4}$$

where $R$ represents the band reflectance. Since OLI has four visible bands, the X, Y, and Z of Landsat OLI data were calculated using the linear weighted summation method as per Chen et al. (2021) (Eqs. (5)–(7)):

$$X = 11.053 \ R(443) + 6.950 \ R(482) + 51.135 \ R(561) + 34.457 \ R(655) \tag{5}$$

$$Y = 1.320 \ R(443) + 21.053 \ R(482) + 66.023 \ R(561) + 18.034 \ R(655) \tag{6}$$

$$Z = 58.038 \ R(443) + 34.931 \ R(482) + 2.606 \ R(561) + 0.016 \ R(655). \tag{7}$$

Once the tristimulus was calculated, the chromaticity coordinates (x, y) were then acquired from the X, Y, and Z (Wang et al., 2021) (Eq. (8)). Afterwards, the hue angle $\alpha$ was derived based on x and y (Van Der Woerd and Wernand, 2018) (Eq. (9)). However, because of the band settings of sensors, there is an offset ($\Delta\alpha$) of the sensor-derived hue angle (Van Der Woerd and Wernand, 2015). Following the ideas in Van Der Woerd and Wernand (2015), Wang (2018) recently developed polynomial deviation delta corrections for multiple sensors (Eq. (10)). Accordingly, the angle $\alpha$ was finally corrected using $\alpha + \Delta\alpha$.

$$x = X/(X + Y + Z), \qquad y = Y/(X + Y + Z) \tag{8}$$

$$\alpha = \mathrm{ARCTAN2}((y - \tfrac{1}{3})/(x - \tfrac{1}{3})) * 180/\pi \tag{9}$$

$$\Delta\alpha = a(\alpha/100)^5 + b(\alpha/100)^4 + c(\alpha/100)^3 + d(\alpha/100)^2 + e\left(\frac{\alpha}{100}\right) + f, \tag{10}$$

where a–f are coefficients of the deviation delta correction, and the correction coefficients of OLI/ETM+/TM are shown in Table 1.

**Table 1.** Polynomial coefficients for the Landsat-OLI/TM/ETM+ hue angle correction (Wang, 2018).

| Sensor | a | b | c | d | e | f |
|---|---|---|---|---|---|---|
| Landsat-TM | 25.851 | -177.4 | 476.69 | -653.3 | 463.33 | -94.41 |
| Landsat-ETM+ | 30.473 | -203.4 | 498.8 | -570.9 | 324.73 | -56.72 |
| Landsat-OLI | 21.355 | -199.29 | 703.3 | -1132.2 | 801.6 | -201.34 |

(2) *Calculation of the FUI*: The FUI for pixels in Landsat were derived from the corrected angle $\alpha$ based on the FUI lookup table (Novoa et al., 2013) (Table 2). Each FUI corresponds to a range of angle $\alpha$.

**Table 2.** The 21-class FUI indices and the corresponding range of hue angle $\alpha$ (Wang et al., 2018; Chen et al., 2021).

| FUI | $\alpha$ range (°) | FUI | $\alpha$ range (°) | FUI | $\alpha$ range (°) |
|---|---|---|---|---|---|
| 1 | (35.00, 42.83) | 8 | (160.97, 175.98) | 15 | (219.34, 224.87) |
| 2 | (42.83, 49.02) | 9 | (175.98, 186.67) | 16 | (224.87, 230.23) |
| 3 | (49.02, 60.01) | 10 | (186.67, 195.44)) | 17 | (230.23, 235.09) |
| 4 | (60.01, 79.23) | 11 | (195.44, 202.05) | 18 | (235.09, 239.56) |
| 5 | (79.23 106.94) | 12 | (202.05, 207.82) | 19 | (239.56, 243.66) |
| 6 | (106.94, 137.03) | 13 | (207.82, 213.57) | 20 | (243.66, 247.25) |
| 7 | (137.03, 160.97) | 14 | (213.57, 219.34) | 21 | (247.25, 252.00) |

(3) *Calculation of the SD*: Based on the calculated FUI and $\alpha$, the SD was obtained following the algorithm proposed in Wang et al. (2020) (Eqs. (11)–(12)). This model had been proved to be suitable for large-area and long-term SD monitoring (Wang et al., 2020).

$$\mathrm{FUI} < 8, \mathrm{SD} = 794630.86 \cdot \alpha^{-1.66} \tag{11}$$





$$\text{FUI} \geq 8, \text{SD} = 30380 \cdot \text{FUI}^{-2.621} \tag{12}$$

*3.3. Assessment of long-term SD trends in China's Lakes*

In order to comprehensively evaluate the long-term trends in SD of natural lakes across China, lakes with areas > 0.01 km² (more than 10 pixels) were manually extracted by referring to the Chinese lake inventories (Chen et al., 2021; Song et al., 2020), the Chinese reservoir and dam database (Song et al., 2018), and high-resolution images from Google Earth. Since previous water investigations were mainly based on MODIS and Sentinel-3 images, and focused on lakes with an area > 1 km², the knowledge of lakes < 1 km² was limited (Zhang et al., 2021b; Chen et al., 2021). Therefore, in our study, the extracted lakes were divided into two groups, lakes with an area > 1 km² and lakes with an area ≤ 1 km², to explore the SD trends in the two different areas. In order to reduce the impact of aquatic vegetation, algae bloom areas, and shallow nearshore on the lake SD assessment, the mean SD of each lake was calculated following the method in Chen et al. (2021). Specifically, the floating algae index (FAI) (Eq. (13)) (Dai et al., 2021; Hu, 2009) was first used to mask algae bloom and aquatic vegetation areas with a threshold of –0.02 (Chen et al., 2021). Then, shallow near-shore pixels were excluded by setting the corresponding threshold value for each lake (Chen et al., 2021). Pixels whose SD was less than the SD value of 80% of the pixels in a given lake were regarded as shallow near-shore pixels and excluded. After the above steps, the remaining pixels in each lake region were used to calculate the mean SD of that lake as follows:

$$\text{FAI} = NIR - (Red + (SWIR - Red) \times (\lambda_{NIR} - \lambda_{Red})/(\lambda_{SWIR} - \lambda_{Red})), \tag{13}$$

where Red, NIR, and SWIR represent the reflectance of red, near-infrared (NIR), and shortwave infrared (SWIR) bands, and $\lambda_{NIR}$, $\lambda_{Red}$, and $\lambda_{SWIR}$ are the center wavelengths of NIR, red, and SWIR bands.

The nonparametric Loess regression method (Steyerberg, 2016) was employed to delineate the long-term SD trend for each lake, and the widely used Mann–Kendall (MK) test (Yuan et al., 2018; Kendall, 1990; Mann, 1945) was applied to indicate the monotonicity of the long-term SD trend. Specifically, the MK indicated the monotonic trend by using a standardized MK statistic Z (Yuan et al., 2018). Z > 0 indicated an upward trend, while Z < 0 indicated a downward trend. The indicated trend was regarded as significant only when $P \leq 0.05$ (Li et al., 2020a). Since the reliability of the long-term trend analysis relied on the observation number of time series data (Li et al., 2020a; Wang et al., 2020), only the lakes that existed in at least 10 SD products were retained for our long-term SD assessment. Using the above criteria, a total of 40,973 lakes were used for time series SD analysis.

**4. Results**

*4.1. Accuracy evaluation of the 30 m long-term LAWSD30 dataset*

The accuracy of our LAWSD30 dataset was evaluated with the collected concurrent in situ SD datasets, as illustrated in Fig. 4. From Fig. 4a, our LAWSD30 dataset exhibited a significant correlation with all collected in situ SD data, with an $R^2$ of 0.86 and an *RMSE* of 0.225 m. Most of the scatter points were distributed close to the 1:1 line. Specifically, from the validation results in the 2010s, our LAWSD30 showed good performance, with an $R^2$ of 0.92 and an *RMSE* of 0.211

m. In addition, a stable performance was also shown in the results for the 2000s, with $R^2$ reaching
0.78 and *RMSE* reaching 0.236 m. Furthermore, a good performance was also seen before the
2000s, with an $R^2$ of 0.69 and an *RMSE* of 0.059 m in the 1990s. The validation results for these
different decades proved the stable performance of our LAWSD30 in different periods. It is
concluded, therefore, that our LAWSD30 can be a reliable dataset for the long-term SD trend
assessment of lakes in China.

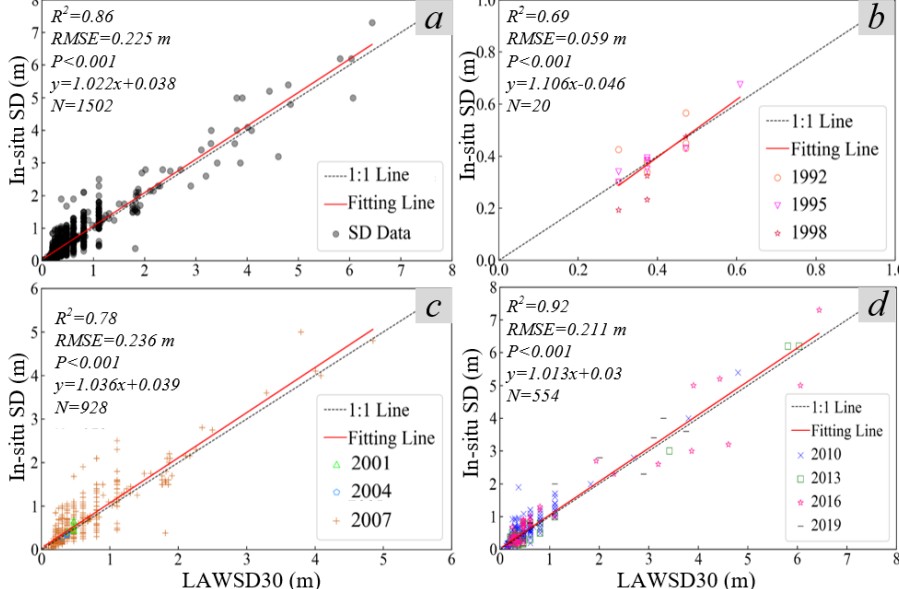


**Figure 4.** Scatterplots of the in situ measured SD data and our LAWSD30 dataset. (**a**) An overall scatterplot
of our LAWSD30 dataset and all the collected in-situ SD data; (**b–d**) scatterplots of our LAWSD30 dataset
and the corresponding in situ SD data in the 1990s (1992, 1995, and 1998), 2000s (2001, 2004, and 2007), and
2010s (2010, 2013, 2016 and 2019), respectively.
*4.2. The LAWSD30 dataset in China*
Our developed long-term LAWSD30 dataset includes 12 SD products of the corresponding
nominal years, now available at https://doi.org/10.5281/zenodo.5734071. Here, the SD product
in 2019 is shown in Fig. 5. It can be found that the water bodies in our product showed a wide
range of SD values (0.1 m to more than 9 m), indicating a great diversity of Chinese inland waters.
Taking the famous Hu line (Hu, 1990) as the boundary, the SD of water bodies showed an
obvious pattern of "high west and low east" across China. The average SD of the water bodies
to the west of the Hu line was approximately 1.7 m, while the average SD of the eastern water
bodies was about 0.4 m. Furthermore, regarding the famous Qinlin–Huaihe line (Liu et al., 2015),
the dividing line between the north and south of China, a significant latitudinal pattern of "high
in the south and low in the north" was exhibited across China. The average SD of the water
bodies distributed to the north of the Qinlin–Huaihe line was about 0.72, whereas the average
SD reached about 1.16 for the water bodies south of this line. The above SD patterns observed
across China were in good agreement with other studies (Wang et al., 2020; Zhang et al., 2021b).

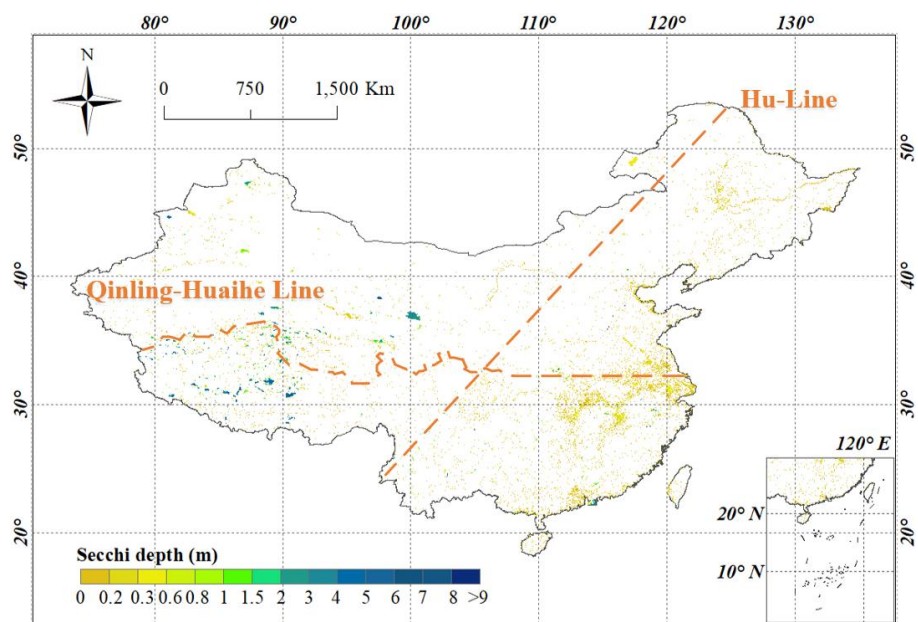


**Figure 5.** The 30 m SD product in China in 2019. Note: the north–south dashed line is the Hu line, and the
east–west dashed line is the Qinling–Huaihe Line.
Moreover, in order to illustrate the ability of our long-term LAWSD30 dataset to monitor
the spatiotemporal pattern of SD in water bodies, the time-series SD results for two important
lakes, Selinco Lake and Hongze Lake, are displayed and used as case studies (Fig. 6). The long-
term mean SD of the two lakes is shown in Fig. 7. From the perspective of the spatial pattern, it
can be seen that the water area in the northern part of Selinco Lake has been increasing, and the
SD of the north is generally lower than that of the central and southern areas of the lake. A
significant pattern of "high center and low north" was found from the SD results for Selinco
Lake. Additionally, an obvious clarity gradient with high values on the northern side and lower
values on the central area and southern side could be found in Hongze Lake. These results are
in good agreement with previous researches (Wang et al., 2020; Xue et al., 2019; Liu et al., 2021).
Furthermore, we can see that the clarity of water in the northern part of Selinco Lake has
improved in recent years, and the SD in the central and southern regions of Hongze Lake has
also increased compared with 35 years ago (Fig. 6). Moreover, in terms of the SD trends, the
mean SD of Selinco Lake exhibited a decreasing but insignificant trend ($Z < 0$, $P > 0.05$) in the
period 1985–2020, while Hongze Lake has shown a significant, increasing SD trend ($Z > 0$, $P <$
0.05) over the past 35 years (Fig. 7). Specifically, the SD curve of Selinco Lake first showed an
upward trend before the 2000s, and then exhibited a decreasing trend after 2001. As for Hongze
Lake, it was found to have an increasing SD trend before 2010, but the SD began to decrease
after that. Similarly, some studies found the same SD change patterns in the two lakes (Liu et
al., 2021; Li et al., 2016; Wang et al., 2020; Zhigang et al., 2017; Li et al., 2019). Therefore, our
long-term SD dataset can provide an opportunity to quickly evaluate the temporal dynamics of
SD in water bodies at low cost, which is of great significance for large-scale water quality



monitoring.

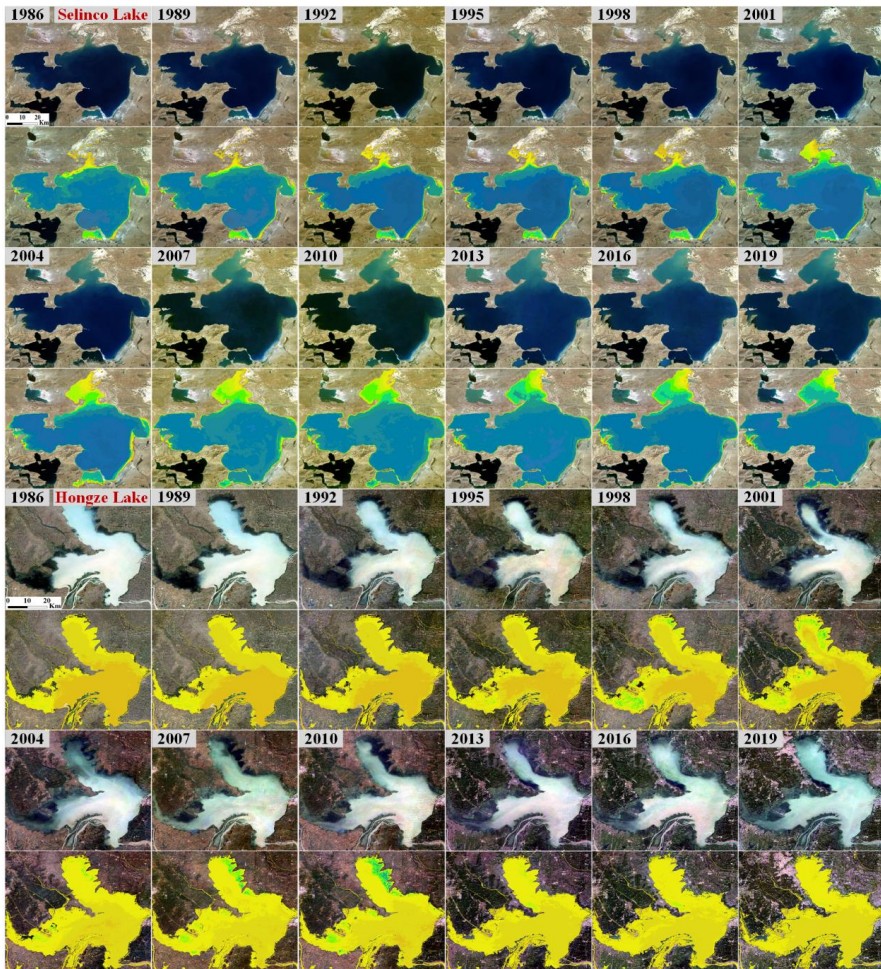

**Figure 6.** The long-term SD results of Selinco Lake and Hongze Lake between 1985 and 2020. Note: the

355                          color bar is the same as that in Fig. 5.

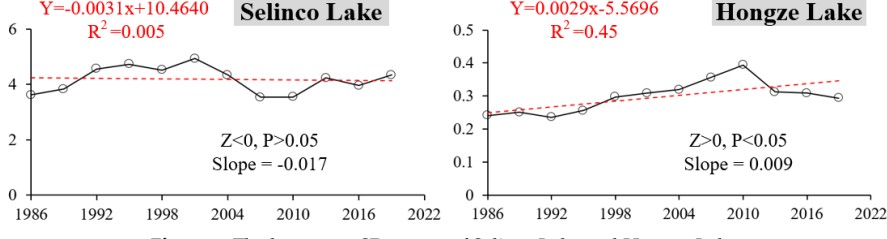

**Figure 7.** The long-term SD curves of Selinco Lake and Hongze Lake.
*4.3. Long-term SD trend of lakes across China in the period 1985–2020*

359          The long-term variations in SD for lakes with an area > 0.01 km² ($N$ = 40,973) across China

from 1985 to 2020 were first evaluated and recognized using our LAWSD30 products (Fig. 8).
First, for lakes with an area ≤ 1 km² (Fig. 8a), the mean SD of the lakes showed a significant



downward trend since 1985 ($Z < 0$, $P < 0.05$), with a rate of –0.055 m/year. However, it can be
seen that the decline rate of SD began to slow down and stabilized after 2001 (with a rate of –
0.026 m/year after 2001). In addition, regarding lakes with areas > 1 km² (Fig. 8b), the mean SD
of the lakes around 2020 was basically the same as that in 1985, and the long-term SD of the
lakes has not shown a significant downward trend since 1985 ($Z < 0$, $P > 0.05$). However, by
carefully observing the time series SD curve of these lakes (Fig. 8b), an obvious turning point
could be found around 2001. The SD of the lakes showed a significant downward trend ($Z < 0$,
$P < 0.05$) before 2001, and began to increase significantly ($Z > 0$, $P < 0.05$) afterward. The above
results demonstrate that the water clarity of lakes in China has continued to improve since the
21st century, but the SD of lakes with an area ≤ 1 km² is still low.

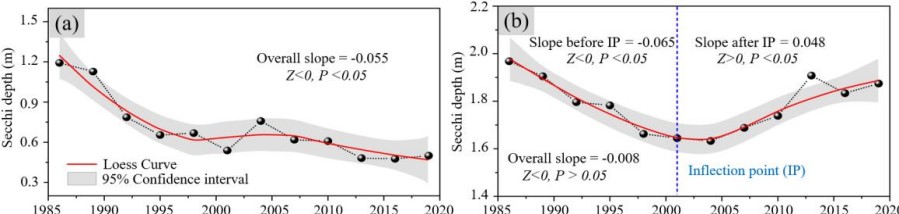


**Figure 8.** The long-term trend in SD for lakes with an area > 0.01 km² ($N = 40,973$) across China from 1985
to 2020. (**a**) The mean SD of lakes ≤ 1 km² across China; (**b**) the mean SD of lakes > 1 km² across China. Note:
IP., inflection point.

376         In order to further evaluate the long-term SD trends of lakes in different geographic regions,
China was divided into five limnetic regions (Ma et al., 2011; Chen et al., 2021), i.e., the Northeast
Mountain Plain Region (NER), Eastern Plain Region (EPR), Yunnan–Guizhou Plateau Region
(YGR), Qinghai–Tibet Plateau Region (QTR), and Mongolian–Xinjiang Plateau Region (MXR)
(Fig. 9). The statistics of lakes with an area > 1 km² and an area ≤ 1 km² are shown in Fig. 9b and
Fig. 9c, respectively. It can be seen that the number of lakes with an area ≤ 1 km² in each region
was far greater than that of lakes with an area > 1 km², but their accumulation area was much
smaller than that of lakes with an area > 1 km². Furthermore, the number and area of lakes in
QTR were the highest, while those in YGR were the lowest.

385         Fig. 10 gives the long-term SD trend of lakes in each limnetic region. For lakes ≤ 1 km² (Fig.
10a–e), it can be seen that, except for MXR and QTR, the SD of the lakes in other regions did not
show a significant decreasing trend ($P > 0.05$) during the entire analysis period. Moreover, the
SD of small lakes (area ≤ 1 km²) in EPR and NER showed obvious increases ($Z > 0$, $P < 0.05$) since
1985, with average change rates of 0.015 m/year and 0.005 m/year, respectively. Although the
SD of small lakes in MXR and QTR experienced significant downward trends over the past 35
years, the decline rates slowed down after the beginning of the 21st century (with rates of 0.001
m/year in MXR and –0.045 m/year in QTR after 2001), and the decrease trend had not been
significant since 2001 ($Z < 0$, $P > 0.05$). Secondly, as for lakes > 1 km², there were no dramatic
decreases in SD in any of the five regions from 1985 to 2020. Moreover, the lakes with an area >
1 km² in NER experienced a significant upward trend in water clarity ($Z > 0$, $P < 0.05$) over the
past 35 years. Additionally, we can also see that the water clarity of lakes > 1 km² in MXR and
QTR significantly improved since the beginning of the 21st century. The SD of lakes > 1 km² in
YGR in 2020 was also higher than that in 1985. However, it should be noted that, although the
SD of lakes > 1 km² in 2020 was also greater than that in 1985 in EPR, the SD of these lakes was
characterized by a significant decrease, with a rate of –0.021 m/year after 2001.

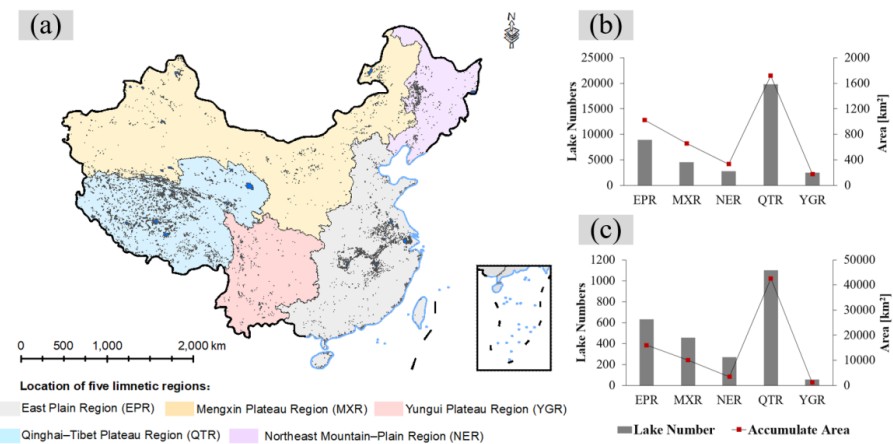


**Figure 9.** The location of the five limnetic regions and the lake statistics in each limnetic region. (**a**) The
location of the five limnetic regions; (**b**) statistics of lakes with areas ≤ 1 km² in each region; (**c**) statistics of
lakes with areas > 1 km² in each region.

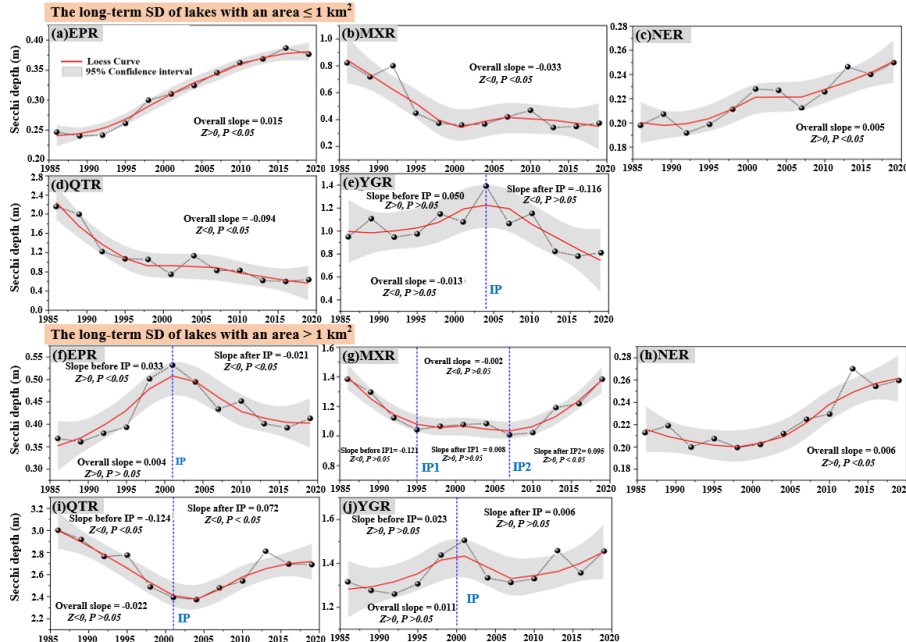


**Figure 10.** The long-term SD trend of lakes in each limnetic region from 1985 to 2020. (**a–e**) The long-term
SD trend of lakes with an area ≤ 1 km² in each region; (**f–j**) the long-term SD trend of lakes with an area >
1 km² in each region. Note: EPR., Eastern Plain Region; MXR., Mongolian–Xinjiang Plateau Region; NER.,
Northeast Mountain Plain Region; QTR., Qinghai–Tibet Plateau Region; YGR., Yunnan–Guizhou Plateau
Region; IP., inflection point.
*4.4 Spatiotemporal patterns of water clarity in lakes over China*





The spatiotemporal patterns of SD in lakes in the five limnetic regions from 1985 to 2020 are
shown in Fig. 11. Overall, for lakes with an area ≤ 1 km² and > 1 km², the average proportions of
lakes with an increasing SD trend were about 76.1% and 81.3%, respectively, in the five limnetic
regions. In addition, the region with the lowest percentage of lakes tending to become clear
(with an increasing trend) was still about 70.0%. The above results indicate that most lakes in
China exhibited a tendency to become clear in the period 1985–2020. Specifically, as for lakes
with areas ≤ 1 km² (hereinafter referred to as small lakes), the minimum proportion of small
lakes whose SD was characterized by an increasing trend was in the MXR (about 70.0%), while
the maximum proportion appeared in the EPR (about 97.4%). In addition, for lakes with areas >
1 km² (hereinafter referred to as large lakes), the smallest and largest proportions of large lakes
that had increasing trends were also in the MXR (about 77.0%) and the EPR (about 84.3%),
respectively.
Focusing on the detailed spatial–temporal SD patterns in each limnetic region, there were
basically no small lakes with a decreasing trend in SD (2.6%) in the EPR. The individual small
lakes that experienced downward trends in EPR were mainly located at the northernmost
regions and at the junction of Hubei and Hunan Provinces. Moreover, as for the large lakes in
the EPR, these were mainly distributed along the Yangtze River, and the lakes showing
decreasing SD trends were mainly located in the middle reaches of the Yangtze River. Secondly,
the MXR was the region with the minimum percentage of lakes that had an increasing trend
among the five limnetic regions. Specifically, small lakes that exhibited decreasing trends were
mainly located in the northeast and northwest of MXR, while large lakes that had decreasing
trends were mainly distributed in the northeast areas of MXR. Additionally, in the NER, large
and small lakes with decreasing SD were mainly distributed in the west and northeast of NER,
accounting for 17.2% of small lakes and 16.1% of large lakes, respectively. Furthermore, in the
QTR, most lakes were located in the north, center, and southeast. Among these lakes, most of
the small lakes with a tendency to become turbid were located in the center, northeast, and
southeast of QTR. In addition, the large lakes that were characterized by decreasing trends were
mainly distributed in the central and northeast parts of the QTR. Lastly, as for the lakes in YGR,
the small lakes that had a decreasing trend were mainly located in the northwest and southeast
of YGR. In contrast, the number of large lakes in the YGR was relatively small ($N$ = 53), and the
lakes with a decreasing trend were mainly distributed in the southeast and west of YGR.
Therefore, although most lakes had a tendency to become clearer from 1985 to 2020, there was
still a considerable proportion of lakes whose SD experienced a downward trend over the past
few decades, which suggests that effective water management is still required in many regions.





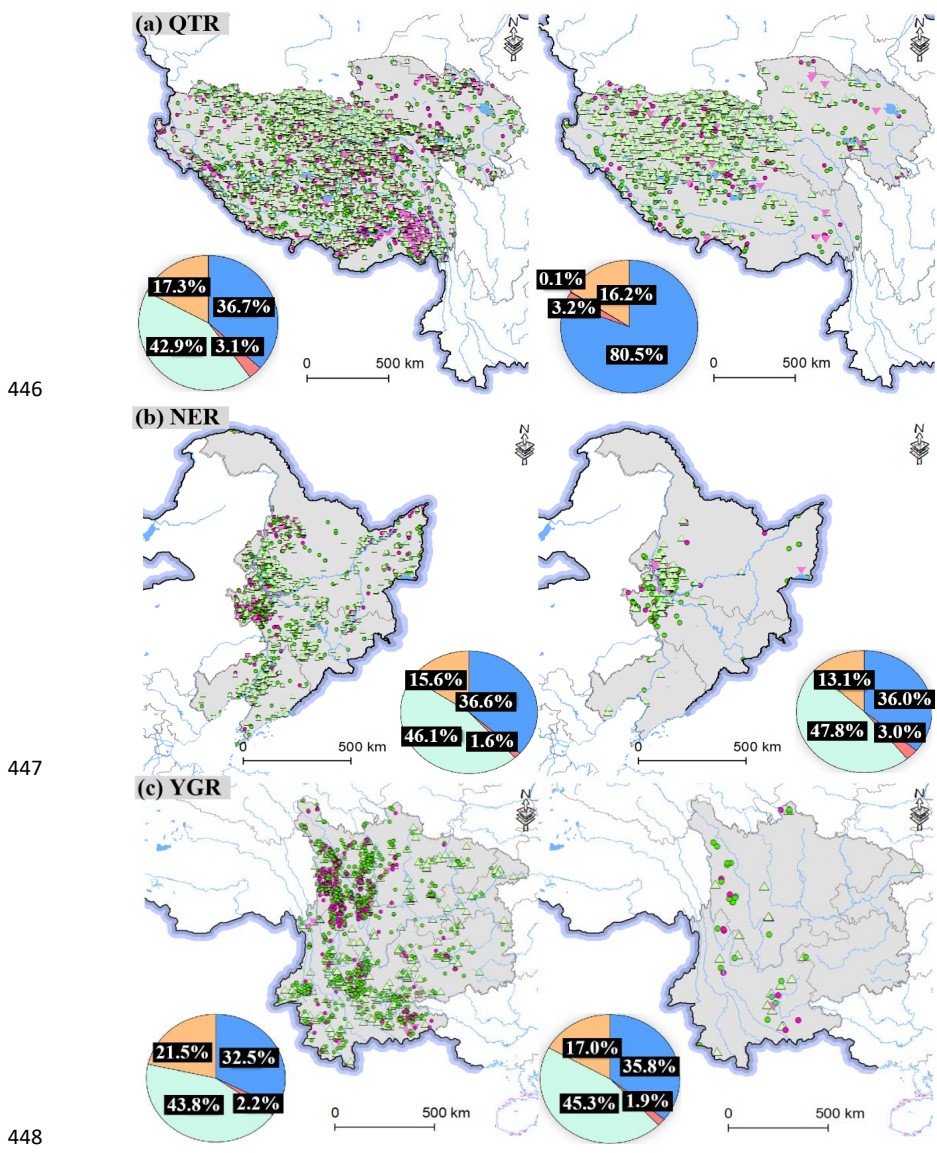






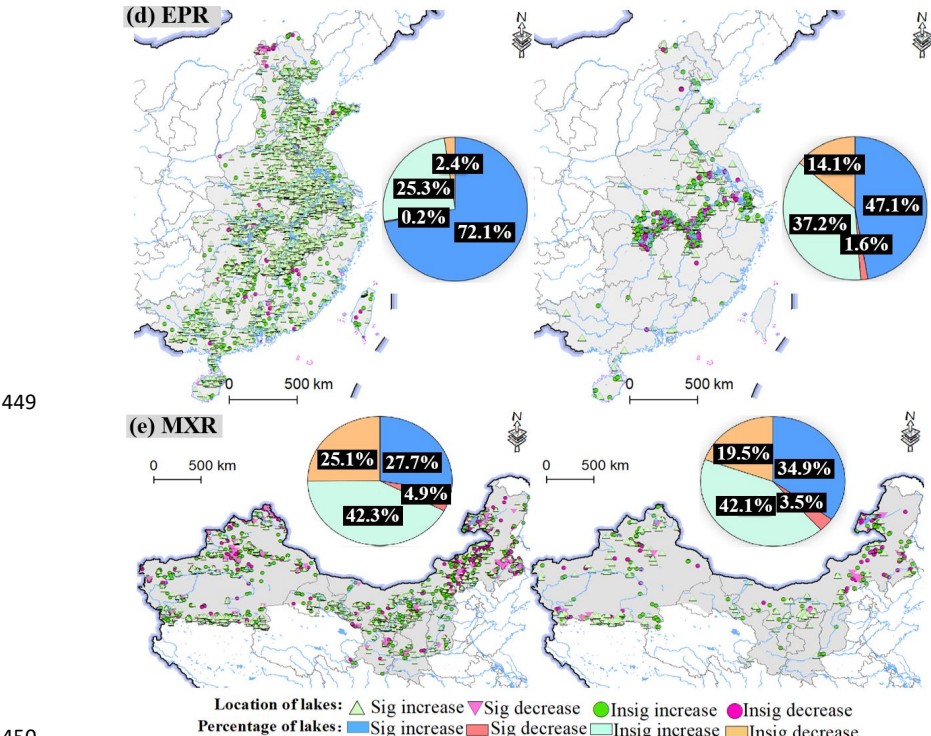

**Figure 11.** The spatiotemporal patterns of SD in lakes in the five limnetic regions from 1985 to 2020 (from
left to right are the spatiotemporal patterns of lakes with an area ≤ 1 km², and the spatiotemporal patterns
of lakes with an area > 1 km²). Note: Sig., significant; Insig., insignificant.
**5. Discussion**
*5.1. Consistency between the Landsat estimation SD results*
Recently, many studies have proved the feasibility of using long-term Landsat series data
from GEE to assess the changes in lake clarity (Zhang et al., 2021b; Yin et al., 2021). In order to
evaluate the comparability of our LAWSD30 dataset in monitoring long-term SD variations, the
Landsat 5, 7, and 8 data for two adjacent tracks with overlapping areas were first selected to test
the consistency between the Landsat estimation SD results (Fig. 12). The images of paths 139
and 140 were chosen because the lakes in this place are hardly affected by human activities, and
thus the SD of lakes can remain stable within a few days under stable hydrometeor conditions
(Zhang et al., 2019b). The Landsat 5 images were taken on October 5, 2011, the Landsat 8 images
were taken on October 21, 2017, and the Landsat 7 ETM + images were taken on October 6, 2011
and October 22, 2017. Since the compared images were quasi-synchronized with each other in
one day, the SD of water bodies was assumed to be the same for both images. Fig. 12c,d show
scatterplots of the SD results for the overlapping regions. It can be seen that, although the model
coefficients of the three sensors were different in our calculation (Section 3.2), there was still
strong consistency between the SD results of Landsat 5, 7, and 8, with an $R^2$ of 0.90 for Landsat
5 vs. 7 and an $R^2$ of 0.97 for Landsat 8 vs. 7. The above results prove that the estimated SD results
from Landsat 5, 7, and 8 data are highly consistent.
Moreover, since SD changes over time, and our LAWSD30 dataset was calculated based on





the BAP composites, it was also necessary to test the phenological consistency between the time-
series summer BAP composites. The mean DOY of each pixel in the BAP composites from 1985
to 2020 was calculated and is shown in Fig. 13a. In addition, the maximum DOY difference of
each pixel location in the BAP composites from 1985 to 2020 was calculated and displayed in
Fig. 13b. From Fig. 13a, most areas of China were composites based on images around DOY 214,
and the mean standard deviation was only 7.5 days. Therefore, the developed BAP composites
can effectively ensure the consistency of phenology between different regions across China. In
addition, from Fig. 13b, the mean value of the maximum DOY difference across China was only
16.5 days, and the maximum DOY differences for most pixels across China (about 94%) were
within 32 days. Although the maximum DOY difference in parts of southern China exceeded 32
days due to the influence of clouds, most of these areas were mountainous with few lakes. In
addition, since the phenology of these regions were in summer, and the SD is relatively stable
during this season (Mccullough et al., 2012; Kloiber et al., 2002), the consistency of water clarity
in these areas can thus be considered not to have much impact on the final result. Therefore, the
results displayed in Figs. 12 and 13 confirm the reliability of our LAWSD30 dataset for
evaluating the long-term SD across China.

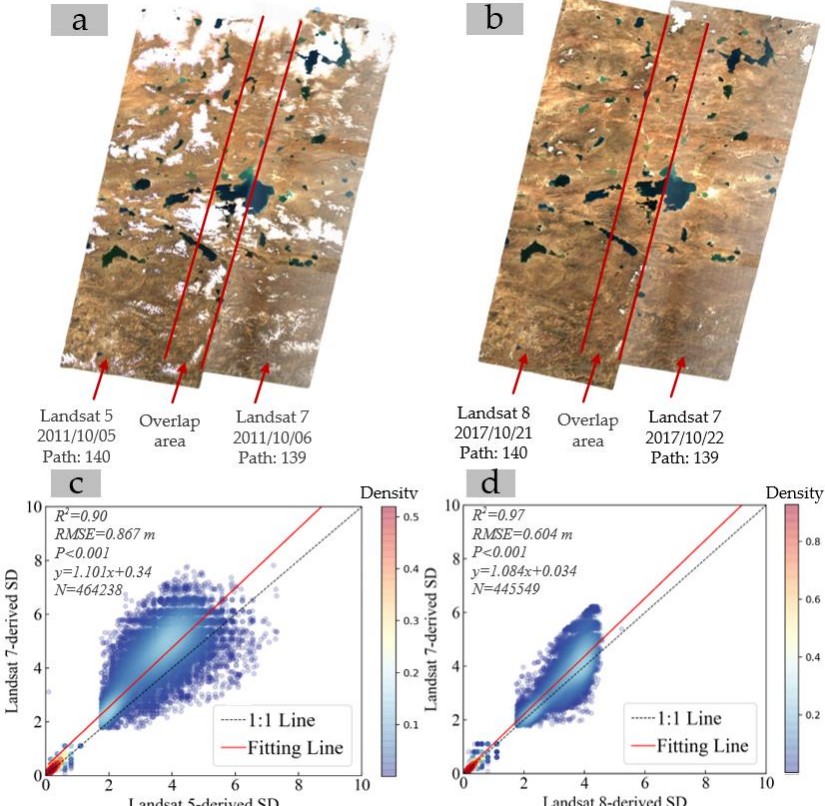

**Figure 12.** Overlapping regions of Landsat 5 and 7 data (**a**) and Landsat 8 and 7 data (**b**); lake SD
comparison for the Landsat 7 vs. Landsat 5 data (**c**) and the Landsat 7 vs. Landsat 8 data (**d**) for the lakes
from the overlap.

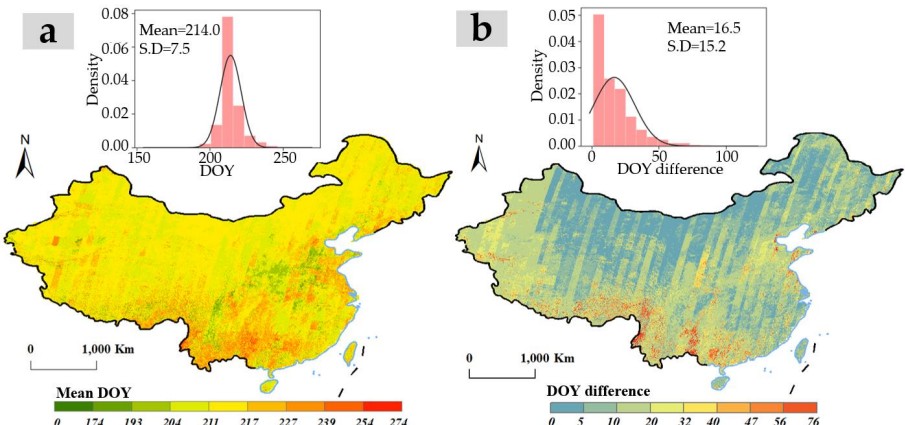

**Figure 13.** The mean DOY of each pixel in the BAP composites from 1985 to 2020 (**a**) and the maximum DOY difference of each pixel location in the BAP composites from 1985 to 2020 (**b**). Note: S.D., standard deviation.

*5.2. Cross-comparison with existing water clarity monitoring studies*

To date, some past studies have also evaluated the water clarity of specific lakes in China (Shen et al., 2020; Wang et al., 2020). In order to further analyze the reliability of our estimated SD results across China, the results of this study were assessed against other existing water clarity monitoring studies. However, since most of the existing investigations focused on the annual average SD (Zhang et al., 2021b; Li et al., 2020a; Yin et al., 2021), and our LAWSD30 dataset is a summer SD dataset, it is a challenge to compare our results with other researches due to the different periods of interest. Fortunately, Wang et al. (2020) recently generated a time-series summer SD dataset (for the period 2000–2017) for lakes > 25 km² in China using water color parameters and MODIS data. Additionally, Shen et al. (2020) developed a multiyear monthly SD dataset (2016-2020) for 86 lakes in eastern China using Sentinel 3 images and a random forest regression SD model. Since both of the studies included SD results in summer, we had a unique opportunity to compare our SD estimates with them. The summer mean SD for each lake in the MODIS and the Sentinel 3-derived SD datasets was calculated and compared with our LAWSD30 dataset. As shown in Fig. 14, our LAWSD30 agreed well with both the MODIS and Sentinel 3-derived SD results. An average $R^2$ of 0.96 and an average $RMSE$ of 0.409 m was achieved when compared with the MODIS-derived results (Fig. 14a,b). In addition, an $R^2$ of 0.74 and an $RMSE$ of 0.109 m were shown in the comparison between the Sentinel 3-derived SD and our LAWSD30 dataset (Fig. 14c). Thus, the above results confirm the reliability of our long-term LAWSD30 dataset.

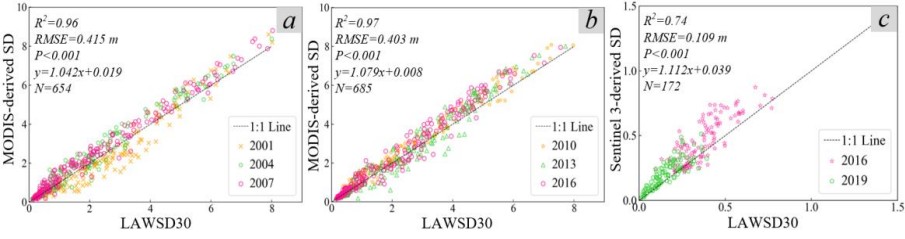





**Figure 14.** (**a**,**b**) Scatterplots of our LAWSD30 data and the corresponding MODIS-derived SD data (Wang et al., 2020) in the 2000s (2001, 2004, and 2007) and 2010s (in 2010, 2013, and 2016), respectively; (**c**) scatterplot of our LAWSD30 data and the corresponding Sentinel 3-derived SD data (Shen et al., 2020) in 2016 and 2019.

## 6. Conclusions

Water clarity is one of the most intuitive and important indicators to reflect the comprehensive conditions in water bodies. In order to improve our understanding of the long-term spatiotemporal patterns of lake water clarity in China, a long-term LAWSD30 dataset with a three-year temporal interval was first developed for the period 1985–2020 using Landsat series data and the GEE platform. The dataset exhibited good performance when compared with concurrent in situ SD measurements (with an $R^2$ of 0.86 and a RMSE of 0.225 m), thus confirming the reliability of our LAWSD30 dataset.

Subsequently, based on the generated LAWSD30 dataset, the national-scale long-term SD estimations of lakes in China ($N$ = 40,973) over the past 35 years were analyzed. It was found that the SD of lakes with an area ≤ 1 km$^2$ showed a significant decreasing trend during the period 1985–2020, but the decline rate began to slow down and stabilized after 2001. Regarding the SD of the lakes with an area > 1 km$^2$, a significant downward trend was seen before 2001, and it began to increase significantly afterwards. In addition, in terms of the spatial patterns, the small lakes showing a decreasing SD trend during 1985–2020 accounted for the largest proportion in MXR (about 30.0%), followed by YGR (23.8%), QTR (20.4%), NER (17.2%), and EPR (2.6%). Additionally, for large lakes, this proportion was the largest in MXR (about 23.0%), followed by QTR (19.4%), YGR (18.9%), EPR (17.7%), and NER (16.1%). The above results indicate that, although the clarity of lakes in China has shown an improving trend since the 21st century, there has still been a considerable proportion of lakes experiencing a downward SD trend over the past few decades. This study can give an effective guidance for the management and restoration of lake water environment.

**Author contributions: Xidong Chen**: Conceptualization, Methodology, Validation, Formal analysis, Investigation, Writing - original draft. **Liangyun Liu**: Conceptualization, Investigation, Writing - review & editing. **Xiao Zhang:** Methodology, Writing - review & editing. **Junsheng Li:** Resources, Writing - review & editing. **Shenglei Wang:** Resources, Validation, Writing - review & editing. **Yuan Gao:** Validation, data curation. **Jun Mi:** Validation, data curation.

**Data availability:** Our long-term LAWSD30 dataset and the lake vector dataset generated for lakes with an area > 0.01 km$^2$ are now available at https://doi.org/10.5281/zenodo.5734071 and https://doi.org/10.5281/zenodo.5734166, respectively. Additionally, our validation datasets can be download at http://lake.geodata.cn.

**Competing interests.** The authors declare that they have no conflict of interest.

**Acknowledgments:** The authors gratefully acknowledge the data support from "National Earth System Science Data Center, National Science & Technology Infrastructure of China (http://www.geodata.cn)", and the financial support provided for this research by the National Natural Science Foundation of China (grant nos. 41825002, 41971318, 41901272) and the Strategic Priority Research Program of the Chinese Academy of Sciences (grant no. XDA19090125).

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

Preliminary Assessment of the Value of Landsat 7 ETM+ Data Following Scan Line Corrector
Malfunction:                    Available                    online                    at:
http://Landsat.usgs.gov/data_products/slc_off_data_products/documents/SLC_off_Scientific_
Usability.pdf. Accessed 12 April 2006, last
van der Woerd, H. J. and Wernand, M. R.: True Colour Classification of Natural Waters with
Medium-Spectral Resolution Satellites: SeaWiFS, MODIS, MERIS and OLCI, Sensors, 15, 25663-
25680, 10.3390/s151025663, 2015.
van der Woerd, H. J. and Wernand, M. R.: Hue-Angle Product for Low to Medium Spatial
Resolution Optical Satellite Sensors, Remote Sensing, 10, 10.3390/rs10020180, 2018.
Wang, S.: Large-scale and Long-time Water Quality Remote Sensing Monitoring over Lakes





Based on Water Color Index, University of Chinese Academy of Sciences, Beijing, 2018.
Wang, S., Li, J., Zhang, W., Cao, C., and Zhang, B.: A dataset of remote-sensed Forel-Ule Index
for global inland waters during 2000–2018, Scientific Data, 8, 10.1038/s41597-021-00807-z, 2021.
Wang, S., Li, J., Zhang, B., Spyrakos, E., Tyler, A. N., Shen, Q., Zhang, F., Kutser, T., Lehmann,
M. K., Wu, Y., and Peng, D.: Trophic state assessment of global inland waters using a MODIS-
derived Forel-Ule index, Remote Sensing of Environment, 217, 444-460, 10.1016/j.rse.2018.08.026,

742    2018.

Wang, S., Li, J., Zhang, B., Lee, Z., Spyrakos, E., Feng, L., Liu, C., Zhao, H., Wu, Y., Zhu, L., Jia,
L., Wan, W., Zhang, F., Shen, Q., Tyler, A. N., and Zhang, X.: Changes of water clarity in large
lakes and reservoirs across China observed from long-term MODIS, Remote Sensing of
Environment, 247, 10.1016/j.rse.2020.111949, 2020.
Wang, X. and Yang, W.: Water quality monitoring and evaluation using remote-sensing
techniques in China: A systematic review, Ecosystem Health and Sustainability, 5, 47-56,

749    10.1080/20964129.2019.1571443, 2019.

White, J. C., Wulder, M. A., Hobart, G. W., Luther, J. E., Hermosilla, T., Griffiths, P., Coops, N.
C., Hall, R. J., Hostert, P., Dyk, A., and Guindon, L.: Pixel-Based Image Compositing for Large-
Area Dense Time Series Applications and Science, Canadian Journal of Remote Sensing, 40, 192-

753    212, 10.1080/07038992.2014.945827, 2014.

Xie, S., Liu, L., Zhang, X., Yang, J., Chen, X., and Gao, Y.: Automatic Land-Cover Mapping using
Landsat Time-Series Data based on Google Earth Engine, Remote Sensing, 11,
10.3390/rs11243023, 2019.
Xue, K., Ma, R., Duan, H., Shen, M., Boss, E., and Cao, Z.: Inversion of inherent optical properties
in optically complex waters using sentinel-3A/OLCI images: A case study using China's three
largest freshwater lakes, Remote Sensing of Environment, 225, 328-346, 10.1016/j.rse.2019.03.006,

760    2019.

Yin, Z., Li, J., Liu, Y., Xie, Y., and Zhang, B.: Water clarity changes in Lake Taihu over 36 years
based on Landsat TM and OLI observations, International Journal of Applied Earth Observation
Geoinformation, 102, 102457, 10.1016/j.jag.2021.102457, 2021.
Yuan, Z., Liang, C., and Li, D.: Urban stormwater management based on an analysis of climate
change: A case study of the Hebei and Guangdong provinces, Landscape and Urban Planning,
177, 217-226, 10.1016/j.landurbplan.2018.04.003, 2018.
Zhang, G., Yao, T., Chen, W., Zheng, G., Shum, C. K., Yang, K., Piao, S., Sheng, Y., Yi, S., Li, J.,
O'Reilly, C. M., Qi, S., Shen, S. S. P., Zhang, H., and Jia, Y.: Regional differences of lake evolution
across China during 1960s-2015 and its natural and anthropogenic causes, Remote Sensing of
Environment, 221, 386-404, 10.1016/j.rse.2018.11.038, 2019a.
Zhang, G. Q., Yao, T. D., Chen, W. F., Zheng, G. X., Shum, C., Yang, K., Piao, S. L., Sheng, Y. W.,
Yi, S., and Li, J. L.: Regional differences of lake evolution across China during 1960s–2015 and
its natural and anthropogenic causes, Remote Sensing of Environment, 221, 386-404,
https://doi.org/10.1016/j.rse.2018.11.038, 2019b.
Zhang, X., Liu, L., Chen, X., Gao, Y., and Jiang, M.: Automatically Monitoring Impervious
Surfaces Using Spectral Generalization and Time Series Landsat Imagery from 1985 to 2020 in
the Yangtze River Delta, Journal of Remote Sensing, 2021, 9873816, 10.34133/2021/9873816, 2021a.
Zhang, X., Liu, L., Wu, C., Chen, X., Gao, Y., Xie, S., and Zhang, B.: Development of a global 30m
impervious surface map using multisource and multitemporal remote sensing datasets with the



Google Earth Engine platform, Earth System Science Data, 12, 1625-1648, 10.5194/essd-12-1625-
781    2020, 2020.
Zhang, Y., Zhang, Y., Shi, K., Zhou, Y., and Li, N.: Remote sensing estimation of water clarity for
various lakes in China, Water Research, 192, 116844, https://doi.org/10.1016/j.watres.2021.116844,
2021b.
Zhigang, C., Hongtao, D., Lian, F., Ronghua, M., and Kun, X.: Climate- and human-induced
changes in suspended particulate matter over Lake Hongze on short and long timescales,
Remote Sensing of Environment, https://doi.org/10.1016/j.rse.2017.02.007, 2017.
Zhou, Q. C., Wang, W. L., Huang, L. C., Zhang, Y. L., Qin, J., Li, K. D., and Chen, L.: Spatial and
temporal variability in water transparency in Yunnan Plateau lakes, China, Aquatic Sciences, 81,
10.1007/s00027-019-0632-5, 2019.