# Peer review of "Long-term water clarity patterns of lakes across China using Landsat series imagery from 1985 to 2020"

_Hydrology and Earth System Sciences, 2021_

## Author Comment (AC1)

Dear Editor,

On behalf of my co-authors, we thank you very much for giving us the opportunity to revise the manuscript (Manuscript ID: **hess-2021-630**). We appreciate the comments on our manuscript entitled "*Long-term water clarity patterns of lakes across China using Landsat series imagery from 1985 to 2020*" by Xidong Chen, Liangyun Liu, Xiao Zhang, Junsheng Li, Shenglei Wang, Yuan Gao, and Jun Mi.

Great thanks to the reviewers and editors, we have revised the manuscript carefully according to the comments. All the changes were high-lighted in the revised manuscript and the point-by-point response to the comments of the reviewers is also listed below.

Please let me know if you require any additional information on our paper.
Looking forward to hearing from you soon.
Best regards,

Xidong Chen

North China University of Water Resources and Electric Power
Zhengzhou 450046, China
Email: chenxd@radi.ac.cn

**Response to Comments from Reviewer 1**
**Reviewer: 1**

This manuscript represents a significant contribution in the form of an **extensive application ready lake clarity data product covering small and large lakes of China over a period of 35 years**. This data product, LAWSD30 (and associated derivations), will be a valuable asset to future endeavors that seek to understand the drivers of lake clarity change in China.

**Response:** Many thanks for the encouraging and positive words.

**Specific comments:**

**Point 1:** Include the normalized RMSE in addition to the RMSE; the normalized RMSE summarizes the explanatory power of a model in a manner that is independent of scale, and so enables a ready comparison across modelling efforts.

**Response:** Many thanks for this comment. According to your comments, we have added the normalized RMSE (NRMSE) and the Mean Absolute Percentage Error (MAPE) to describe the model errors. The corresponding description and figures have been revised, and the details are shown below.

P10 Line 304 – Line 318: 'From Fig. 4a, our LAWSD30 dataset exhibited a significant correlation with all collected in situ SD data, with an $R^2$ of 0.92, a Root Mean Square Error (*NRMSE*) of 28.1%, and a Mean Absolute Percentage Error (*MAPE*) of 0.235. Most of the scatter points were distributed close to the 1:1 line. Specifically, from the validation results in the 2010s, our LAWSD30 showed good performance, with an $R^2$ of 0.92, a *NRMSE* of 30.8%, and a *MAPE* of 0.268. In addition, a stable performance was also shown in the results for the 2000s, with $R^2$ reaching 0.84, *NRMSE* reaching 30.1%, and MAPE reaching 0.219. Furthermore, a good performance was also seen before the 2000s, with an $R^2$ of 0.69, a *NRMSE* of 15.0%, and a *MAPE* of 0.126 in the 1990s. The validation results for these different decades demonstrated the stable performance of our LAWSD30 in different periods. It is concluded, therefore, that our LAWSD30 can be a reliable dataset for the long-term SD trend assessment of lakes in China.

[Figure]

**Figure 4.** Scatterplots of the in situ measured SD data and our LAWSD30 dataset. (**a**) An overall scatterplot of our LAWSD30 dataset and all the collected in-situ SD data; (**b–d**) scatterplots of our LAWSD30 dataset and the corresponding in situ SD data in the 1990s (1992, 1995, and 1998), 2000s (2001, 2004, and 2007), and 2010s (2010, 2013, 2016 and 2019), respectively.'

P18 Line 485 – P19 Line 490: 'It can be seen that, although the model coefficients of the three sensors were different in our calculation (Section 3.2), there was still strong consistency between the SD results of Landsat 5, 7, and 8, with an $R^2$ of 0.90 for Landsat 5 vs. 7 and an $R^2$ of 0.97 for Landsat 8 vs. 7 (Fig. 12). In addition, the *NRMSE* and *MAPE* for Landsat 5 vs. 7 and Landsat 8 vs. 7 were both lower than 26% and 0.22, respectively. The above results demonstrate that the estimated SD results from Landsat 5, 7, and 8 data are highly consistent.

[Figure]

**Figure 12.** Overlapping regions of Landsat 5 and 7 data (**a**) and Landsat 8 and 7 data (**b**); lake SD comparison for the Landsat 7 vs. Landsat 5 data (**c**) and the Landsat 7 vs. Landsat 8 data (**d**) for the lakes from the overlap.'

P20 Line 531 – Line 535: 'An average $R^2$ of 0.96 was achieved when compared with the MODIS-derived results, and the average *NMRSE* and *MAPE* were only 18.4% and 0.178, respectively (Fig. 14a,b). In addition, an $R^2$ of 0.74, a *NMRSE* of 28.0%, and a *MAPE* of 0.225 were shown in the comparison between the Sentinel 3-derived SD and our LAWSD30 dataset (Fig. 14c).

[Figure]

**Figure 14.** (**a**,**b**) Scatterplots of our LAWSD30 data and the corresponding MODIS-derived SD data (Wang et al., 2020) in the 2000s (2001, 2004, and 2007) and 2010s (in 2010, 2013, and 2016), respectively; (**c**) scatterplot of our LAWSD30 data and the corresponding Sentinel 3-derived SD data (Shen et al., 2020) in 2016 and 2019.'

**Point 2:** Line 118: replace 'product' with 'application ready data product'.

**Response:** Many thanks for this comment. According to your comments, the corresponding description and have been revised, and the details are shown below.

P3 Line 117 – Line 118: 'Taking into account the frequent contamination of cloud and cloud shadow, it is hard to develop a spatially continuous application ready data product throughout China with only one year of Landsat images (Zhang et al., 2019a).'

**Point 3:** Lines 147 - 48: it is not clear how 'user' and 'producer' map to the terminology used in Peckel et al 2016. Please clarify or re-word.

**Response:** Many thanks for this comment. According to your comments, the corresponding description and have been revised. The terms 'commission' and 'omission' accuracies were used to replace the user and producer accuracies, and the details are shown below.

P5 Line 151 – Line 152: 'The overall commission and omission accuracies for permanent water were 99.6% and 98.6%, respectively, versus 98.6% and 75.4% for seasonal water (Pekel et al., 2016).'

**Point 4:** Line 211: replace 'Following him' with 'Following Chen et al. (2021)', assuming that is what is meant?

**Response:** Great thanks for this comment. According to your comments, the corresponding description and have been revised. The details are shown below.

P7 Line 216 – Line 218: 'Following Chen et al. (2021) , the DOY criteria (Eq. (A1)), the cloud and cloud shadow criteria (Eq. (A2)), and the atmospheric opacity criteria (Eq. (A3)) were selected to generate the BAP composites.'

**Point 5:** Line 215: for the convenience of the reader, please briefly describe the constituent BAP criteria and how the BAP was calculated, rather than pointing to Chen et al. 2021 (which then on-refers the reader to White et al. 2014 for further details).

**Response:** Great thanks for this comment. According to your comments, we have added an Appendices to briefly describe the details of the BAP criteria. Also, the corresponding description in the text were revised. The details are shown below.

P7 Line 216 – Line 219: 'Following Chen et al. (2021), the DOY criteria (Eq. (A1)), the cloud and cloud shadow criteria (Eq. (A2)), and the atmospheric opacity criteria (Eq. (A3)) were selected to generate the BAP composites (the details of the criteria are presented in the Appendices).'

P22 Line 582 – Line 592: '

**7. Appendices**

In order to generate the BAP composites, three criteria (i.e., the DOY criteria (Eq. (A1)), the cloud and cloud shadow criteria (Eq. (A2)), and the atmospheric opacity criteria (Eq. (A3)) were selected:

$$\text{Score}_{DOY} = \frac{1}{\sigma\sqrt{2\pi}} e^{-\frac{1}{2}(\frac{x_j-\mu}{\sigma})^2} \tag{A1}$$

$$\text{Score}_{CloudDisctance} = \frac{1}{1 + e^{(-0.2(\min(D_i, D_{req})-\frac{D_{req}}{2}))}} \tag{A2}$$

$$Score_{Opacity} = 1 - \frac{1}{1 + e^{(-0.2(\min(O_i, O_{max}) - (\frac{O_{max} - O_{min}}{2})))}} \tag{A3}$$

where $\mu$ and $\sigma$ represent the target DOY and the standard deviation of the DOY, respectively, and $x_j$ is the DOY for image $j$. The target DOY $\mu$ and the standard deviation of the DOY was set to August 1 and 60, respectively. $D_i$ is the distance from pixel $i$ to cloud or cloud shadow, and $D_{req}$ is the minimum required distance, which was set to 50 here. $O_i$ is the opacity of pixel $i$, while $O_{max}$ and $O_{min}$ are the minimum opacity value, respectively. $O_{max}$ was set to 0.3 and $O_{min}$ was set to 0.2. All parameters for the three criteria were set according to Chen et al. (2021). The BAP score was then derived as followed:

$$Score_{BAP} = Score_{DOY} + Score_{CloudDisctance} + Score_{Opacity} \tag{A4}$$

**Point 6:** Line 227: please replace the word 'proven' with 'demonstrated', here and elsewhere in the document. Likewise, replace 'proves' with 'demonstrates'. 'Proven' and 'proves' should be reserved for contexts where there is little room for interpretation.

**Response:** Great thanks for this comment. According to your comments, we have checked the full text and revised the corresponding descriptions. The details are shown below.

P8 Line 233 – Line 234: 'Previous researches have demonstrated that FUI and hue angle ($\alpha$) are useful water color parameters for assessing the SD of inland waters.'

P9 Line 268 – Line 269: 'This model had been demonstrated to be suitable for large-area and long-term SD monitoring.'

P18 Line 473 – Line 474: 'Recently, many studies have demonstrated the feasibility of using long-term Landsat series data from GEE to assess the changes in lake clarity.'

P18 Line 489 – P19 Line 490: 'The above results demonstrate that the estimated SD results from Landsat 5, 7, and 8 data are highly consistent.'

**Point 7:** Line 316: replace 'product' with 'LAWSD30 dataset'.

**Response:** Great thanks for this comment. According to your comments, we have revised the corresponding descriptions. The details are shown below.

P11 Line 327 – Line 328: 'It can be found that the water bodies in our LAWSD30 dataset product showed a wide range of SD values.'

**Point 8:** Line 339: replace 'researches' with 'studies' (here and elsewhere in the manuscript).

**Response:** Great thanks for this comment. According to your comments, we have checked the full text and revised the corresponding descriptions. The details are shown below.

P12 Line 355 – Line 356: 'These results are in good agreement with previous studies.'

P20 Line 522 – Line 523: 'it is a challenge to compare our results with other studies due to the different periods of interest.'

**Point 9:** Line 348: replace 'Similarly, some studies found the same SD ....' with 'Other studies found the same SD ...'.

**Response:** Great thanks for this comment. According to your comments, we have revised the corresponding descriptions. The details are shown below.

P12 Line 365: 'Similarly, other studies found the same SD change patterns in the two lakes.'

**Point 10:** Line 462: replace 'hydrometeor' with 'weather'. (hydrometeor has a different meaning to that which I assume is intended).

**Response:** Great thanks for this comment. According to your comments, we have revised the corresponding descriptions. The details are shown below.

P18 Line 479 – Line 480: 'thus the SD of lakes can remain stable within a few days under stable weather conditions.'

**Point 11:** Lines 523 - 24: replace 'to reflect the comprehensive conditions in water bodies' with 'of water quality'.

**Response:** Great thanks for this comment. According to your comments, we have revised the corresponding descriptions. The details are shown below.

P21 Line 561– Line 562: 'Water clarity is one of the most intuitive and important indicators of water quality.'

**Point 12:** Conclusion: include commentary that alerts the reader to the notion that an informed interpretation of the lake clarity patterns presented in this manuscript will require an understanding of the relative importance of local and regional driving forces; and that such an informed interpretation will be required in order to prioritise management actions.

For example, the work of Yu and Zhai (https://doi.org/10.1038/s41598-020-71312-3) demonstrate that compound drought and heat events have impacted China more severly since ca. the year 2000. Have such events led to changes in SD patterns in some lakes, for example by making wind-driven resuspension more or less likely?

**Response:** Great thanks for this comment. According to your comments, we conducted an experiment to briefly analyze the potential linkage between environmental factors and the lake clarity in China. The details are shown below:

In order to preliminarily explore the potential linkage between environmental factors and the spatial-temporal changes of lake clarity in the five limnetic regions, three categories of environmental factors were collected, including: lake area factors (i.e. lake area (LA) of lakes with areas ≤ 1 km² and the LA of lakes with areas > 1 km²), meteorological factors (i.e. precipitation (PR), wind speed (WS), and near surface air temperature (AT)), and human activity factors (i.e. local gross domestic production (GDP) and population density (PO)). These factors have been considered as potential drivers of the long-term SD trend of lakes in previous studies (Feng et al. 2019a; Li et al. 2020a; Meng et al. 2015; Wang et al. 2020; Yin et al. 2021).

The collected environmental factors were separately counted in each region. Specifically, the annual summer mean values for the PR, WS, and AT in each limnetic region were calculated. Also, the total LA and the intra-year PO and GDP of each limnetic region was calculated for each target year. The long-term variations in these environmental factors were shown in Fig. R1. The GDP, PO, PR, and LA exhibited increase trends in all five limnetic regions from 1985 to 2020. In addition, WS was characterized by increase trends in MXR, YGR, and NER, but downward trends in EPR and QTR. Moreover, AT has been trending upward in EPR, MXR, QTR, and YGR since 1985, but has decreased in NER. The correlation between these factors and

the variations in lake SD was analyzed and shown in Fig. R2.

Regarding the correlations in the east (including EPR and NER), the human activity factors (PO and GDP) showed strong positive correlations with small lakes (with areas ≤ 1 km²) (P < 0.01). Also, a strong positive correlation between GDP and clarity of large lakes (with areas > 1 km²) can be found in NER. In fact, with the development of GDP and urbanization, local and national governments have put forward a variety of regulations to manage water resource and control pollution (Jiang and Zheng 2011; Wang et al. 2020). Our results appeared to further proved the effectiveness of the environmental regulation and projects implemented by the local and national governments. Similarly, previous works also found PO and GDP were positively correlated with the lake clarity in eastern China (Feng et al. 2019a; Guo et al. 2015; Zhao et al. 2018). In addition, the correlation between large lakes and human activity factors was found to be relatively weak in EPR (the correlation coefficient (r) between SD and PO and GDP was 0.12, and -0.09, respectively), which means the regulation and management for small lakes may be stronger than that for large lakes. Furthermore, the small lakes in EPR also showed significant positive correlations with the increase of AT and LA, indicating the positive impact of the two factors on SD of small lakes in EPR. Nevertheless, since warm AT generally promotes the growth of algae, causing algae blooms and reducing water clarity (Li et al. 2020a; Yin et al. 2021), it seems that SD should not increase with the increase in AT. However, because climate, lake area, and human activities are interconnected in complex ways (Maberly and Elliott 2012; Williamson et al. 2009), and other factors (such as GDP, PO, and LA) are also significantly positively correlated with SD, it is possible to lead a positive correlation between AT and SD. Similarly, Wang et al. (2020) and Feng et al. (2019a) also found that some lakes in EPR were positively correlated with lake SD. Then, for the large lakes in EPR, WS and SA showed significant negative and positive correlations with lake clarity, respectively, implying the negative and positive effects of WS and SA on SD in large lakes in this region. Similar results can also be found in recent studies (Feng et al. 2019a; Xue et al. 2019).

In MXR, PO and AT showed significant negative correlations with SD changes in small lakes, suggesting the negative effects of the two factors on SD variations in small lakes. In contrast, all factors exhibited weak and insignificant correlations with large lakes in EPR, implying that large lakes in MXR are more resistant to environmental changes than small lakes.

As for lakes in QTR, the small lakes had significant negative correlations with PO, GDP, AT, and LA, suggesting the negative effects of the four factors on the long-term SD changes in small lakes from 1985 to 2020. Several studies have reported that climate warming has accelerated the melting of glaciers and permafrost in cold regions over recent decades, leading to an increase in the lake area and water volume in QTR (Liu et al. 2021; Sun et al. 2018; Yang et al. 2017; Zhang et al. 2011). Since the sediments carried by meltwater can reduce the SD in lakes (Mi et al. 2011), this is probably the reason for the negative correlations between SD and AT and LA. In addition, because high AT can also affect the growth of phytoplankton in QTR lakes (Feng et al. 2019b), this may also be one of the reasons for the negative effects of AT. In agreement with our results, recent studies have also found the negative effects of AT and LA on lake SD in QTR (Liu et al. 2021; Wang et al. 2020). In contrast, from the results of large lakes in QTR, the correlations between AT and LA and water clarity were shown to be weak, implying that the impact of climate warming on small lakes is significantly stronger than that of large lakes in QTR. The above results can provide effective support for the protection and

governance of the lake environment in this region.

Lastly, for lakes in YGR, small lakes had significant negative correlations with GDP and WS, while none of a factor showed a significant correlation with large lakes. Since the increase in GDP of the catchment may indirectly increase the input of contaminants and nutrients in water bodies (Zhou et al. 2019), and the recovery capacity of small lakes is generally limited (Helen et al. 2015), GDP and the water clarity of small lakes are therefore likely to be negatively correlated. Recently, Xu et al. (2018) found similar correlations between GDP and SD in lakes in YGR. In addition, because wind speed can accelerate the growth of cyanobacteria, resulting in a decrease in the SD of the lake (Shi et al. 2018; Yin et al. 2021), the negative correlation between WS and SD can be well understood and associated with this.

[Figure]

**Figure R1.** Long-term curves of the human activity factors, meteorological factors, and lake area factors in the five limnetic regions (EPR, MXR, QTR, YGR, and NER) from 1985 to 2020. *Noted:* LA., lake area; PR., precipitation; WS., wind speed; AT., near surface air temperature; GDP., gross domestic production; PO., population density.

| | EPR | | MXR | | QTR | | YGR | | NER | |
|---|---|---|---|---|---|---|---|---|---|---|
| | Area ≤ 1 km² | Area > 1 km² | Area ≤ 1 km² | Area > 1 km² | Area ≤ 1 km² | Area > 1 km² | Area ≤ 1 km² | Area > 1 km² | Area ≤ 1 km² | Area > 1 km² |
| PO | 0.97 *** | 0.12 | -0.83 *** | -0.17 | -0.87 *** | -0.42 | 0.11 | 0.48 | 0.79 ** | 0.57 |
| GDP | 0.88 *** | -0.09 | -0.55 | 0.29 | -0.66 * | -0.01 | -0.60 * | 0.44 | 0.88 *** | 0.93 *** |
| PR | 0.34 | -0.28 | -0.18 | 0.18 | -0.51 | -0.50 | -0.01 | 0.08 | 0.22 | 0.23 |
| AT | 0.77 *** | -0.01 | -0.66 * | -0.11 | -0.67 * | -0.32 | -0.48 | 0.19 | 0.36 | 0.41 |
| WS | -0.40 | -0.68 * | -0.49 | 0.38 | 0.21 | -0.06 | -0.62 * | 0.12 | 0.03 | 0.04 |
| LA | 0.68 * | 0.60 * | -0.52 | 0.19 | -0.71 ** | -0.25 | 0.15 | 0.33 | 0.25 | 0.51 |

**Figure R2.** Pearson correlation coefficients (r) between SD and the six environmental factors of lakes with areas > 1 km² and ≤ 1 km² in the five limnetic regions (EPR, MXR, QTR, YGR, and NER).

However, since the focus of our manuscript is on our LAWSD30 dataset and the monitoring of the long-term SD trends of lakes across China, we have not conducted more specific experiments in analyzing the driver factors of SD changes. Also, considering the length of the article and to keep the article focused, we did not incorporate the above analysis into the manuscript. Instead, we added a section "*5.3 limitations and suggestions for future work*" to the Discussion to illustrate the limitations of our study in analyzing the driver factors of SD changes, and to inform readers that investigations into the drivers of SD change are the main focus of our follow-up work. Also, the work of Yu and Zhai (https://doi.org/10.1038/s41598-020-71312-3) was cited in this section to illustrate the possible driving factors that we will analyze. The details are shown below:

P20 Line 543 – P21 Line 559: '

*5.3 limitations and suggestions for future work*

First, due to the relative long revisit period of Landsat series data and the interference from clouds, our study is not suitable for the evaluation of the intra-annual changes in water clarity. The long-term intra-annual and seasonal SD changes of lakes across China still need to be evaluated. In the future, more efforts will be devoted to the use of more multi-source data to increase the number of valid observations, so as to conduct more analysis on SD at seasonal scale.

Second, because this study is mainly focused on monitoring the long-term SD changes in lakes across China, the comprehensive analysis of the relationship between variation of lake water clarity and environmental factors was not investigated. Since the discharge of agricultural fertilizers, industrial sewage, and livestock excrement within the lake drainage area, as well as the spatial-temporal variations of land use and land cover can affect the clarity of the water bodies (Li et al., 2020a), data on related factors should be collected and analyzed. In addition, the topographic conditions of the lake basin, such as the slope of the lake shoreline and the size of recharge river, and the extreme weather and climate events (i.e. drought and heat event) (Liu et al., 2021a; Yu and Zhai, 2020), may also have important impact on the lake clarity. In future work, we will do more investigations to analyze the above potential driver factors of SD changes.'

---

## Author Comment (AC2)

Dear Editor,

On behalf of my co-authors, we thank you very much for giving us the opportunity to revise the manuscript (Manuscript ID: **hess-2021-630**). We appreciate the comments on our manuscript entitled "*Long-term water clarity patterns of lakes across China using Landsat series imagery from 1985 to 2020*" by Xidong Chen, Liangyun Liu, Xiao Zhang, Junsheng Li, Shenglei Wang,Yuan Gao, and Jun Mi.

Great thanks to the reviewers and editors, we have revised the manuscript carefully according to the comments. All the changes were high-lighted in the revised manuscript and the point-by-point response to the comments of the reviewers is also listed below.

Please let me know if you require any additional information on our paper.
Looking forward to hearing from you soon.
Best regards,

Xidong Chen

North China University of Water Resources and Electric Power
Zhengzhou 450046, China
Email: chenxd@radi.ac.cn

**Response to Comments from Reviewer 2**
**Reviewer: 2**

This paper developed a 30 m long-term Lake Water Secchi Depth (SD) dataset (LAWSD30) of China (1985–2020) using the robust water-color-parameter-based SD model. The LAWSD30 dataset can be used to study the temporal and spatial changes of Lake SD. Therefore, it is of great significance to the study of long-term trend of Lake SD and lake ecological environment management.

**Response:** Great thanks for the encouraging and positive words.

**Specific comments:**

**Point 1:** The RMSE was used for evaluating the accuracy of the model. However, in my opinion, this index is not very appropriate. For example, when the real value is equal to 2 or 20, although the RMSE value is 0.2, the accuracy of the model is very different. Thus, this index does not show how close the real value is to the estimated value I suggest using MAPE or similar indicators to evaluate the accuracy of the model.

**Response:** Many thanks for this comment. According to your comments, we have added the normalized RMSE (NRMSE) and the Mean Absolute Percentage Error (MAPE) to describe the model errors. The corresponding description and figures have been revised, and the details are shown below.

P10 Line 304 – Line 318: 'From Fig. 4a, our LAWSD30 dataset exhibited a significant correlation with all collected in situ SD data, with an $R^2$ of 0.92, a Root Mean Square Error (*NRMSE*) of 28.1%, and a Mean Absolute Percentage Error (*MAPE*) of 0.235. Most of the scatter points were distributed close to the 1:1 line. Specifically, from the validation results in the 2010s, our LAWSD30 showed good performance, with an $R^2$ of 0.92, a *NRMSE* of 30.8%, and a *MAPE* of 0.268. In addition, a stable performance was also shown in the results for the 2000s, with $R^2$ reaching 0.84, *NRMSE* reaching 30.1%, and MAPE reaching 0.219. Furthermore, a good performance was also seen before the 2000s, with an $R^2$ of 0.69, a *NRMSE* of 15.0%, and a *MAPE* of 0.126 in the 1990s. The validation results for these different decades demonstrated the stable performance of our LAWSD30 in different periods. It is concluded, therefore, that our LAWSD30 can be a reliable dataset for the long-term SD trend assessment of lakes in China.

[Figure]

**Figure 4.** Scatterplots of the in situ measured SD data and our LAWSD30 dataset. (**a**) An overall scatterplot of our LAWSD30 dataset and all the collected in-situ SD data; (**b–d**) scatterplots of our LAWSD30 dataset and the corresponding in situ SD data in the 1990s (1992, 1995, and 1998), 2000s (2001, 2004, and 2007), and 2010s (2010, 2013, 2016 and 2019), respectively.'

P18 Line 485 – P19 Line 490: 'It can be seen that, although the model coefficients of the three sensors were different in our calculation (Section 3.2), there was still strong consistency between the SD results of Landsat 5, 7, and 8, with an $R^2$ of 0.90 for Landsat 5 vs. 7 and an $R^2$ of 0.97 for Landsat 8 vs. 7 (Fig. 12). In addition, the *NRMSE* and *MAPE* for Landsat 5 vs. 7 and Landsat 8 vs. 7 were both lower than 26% and 0.22, respectively. The above results demonstrate that the estimated SD results from Landsat 5, 7, and 8 data are highly consistent.

[Figure]

**Figure 12.** Overlapping regions of Landsat 5 and 7 data (**a**) and Landsat 8 and 7 data (**b**); lake SD comparison for the Landsat 7 vs. Landsat 5 data (**c**) and the Landsat 7 vs. Landsat 8 data (**d**) for the lakes from the overlap.'

P20 Line 531 – Line 535: 'An average $R^2$ of 0.96 was achieved when compared with the MODIS-derived results, and the average *NMRSE* and *MAPE* were only 18.4% and 0.178, respectively (Fig. 14a,b). In addition, an $R^2$ of 0.74, a *NMRSE* of 28.0%, and a *MAPE* of 0.225 were shown in the comparison between the Sentinel 3-derived SD and our LAWSD30 dataset (Fig. 14c).

[Figure]

**Figure 14.** (**a**,**b**) Scatterplots of our LAWSD30 data and the corresponding MODIS-derived SD data (Wang et al., 2020) in the 2000s (2001, 2004, and 2007) and 2010s (in 2010, 2013, and 2016), respectively; (**c**)

scatterplot of our LAWSD30 data and the corresponding Sentinel 3-derived SD data (Shen et al., 2020) in 2016 and 2019.'

**Point 2:** Line 125-126. "The summer months were chosen because the water clarity is relatively stable in this season and suitable for monitoring with remote sensing imagery". I understand that calculating SD in the same season can enhance the comparability of data, but I don't think the clarity is relatively stable in the summer, because heavy rainfall and algal bloom often occur in summer, resulting in the change of suspended solids and therefore affecting the SD.

**Response:** Many thanks for this comment. We agree that the rainfall and algal bloom in summer may affect the SD of lakes. However, due to the abundant lake water in summer, the lake water is relatively deep in this season, which is suitable for retrieving the clarity of lake bodies. In addition, previous studies demonstrated that the SD of lakes is more stable in summer than in other seasons when there are no climatic events (i.e. heavy rainfall, storm events) (Song et al., 2020; McCullough et al., 2012; Stadelmannet al., 2001). Furthermore, summer is also the period with the greatest abundance of volunteer-collected in-situ water quality data (McCullough et al., 2012). Therefore, summer months were preferable to be used for remote sensing-based water clarity estimating (Chen et al., 2021; Wang et al., 2020; Song et al., 2020). Also, in the process of water clarity mapping, we have utilized the time series NDTI to mitigate the disturbance of the climatic events on water clarity:

P7 Line 222 – P8 Line 229: 'However, since floods and rainfall in summer will bring suspended particles into water bodies, making the SD of water bodies much lower than usual (Murshed et al., 2014; Liu et al., 2021a), it is also necessary to reduce the impact of these climatic events on the BAP composites to ensure the reliability of the long-term SD trend assessment. Here, the normalized difference turbidity index (NDTI) (Lacaux et al., 2007) was used to indicate the turbidity of the water (Eq. (1)). As the SD of water decreases, the NDTI of water increases (Islam, 2006; Lacaux et al., 2007). Therefore, the interference of floods and rainfall was restricted by only using the observations with NDTI less than the 80[th] quantile of their NDTI stack for BAP compositing.

$$NDTI = (Red - Green)/(Red + Green) \qquad (1)$$ '

*Song, K., Liu, G., Wang, Q., Wen, Z., Lyu, L., Du, Y., Sha, L., Fang, C.: Quantification of lake clarity in China using Landsat OLI imagery data, Remote Sensing of Environment, 243, 2020.*

*McCullough, I. M., Loftin, C. S., Sader, S. A.: Combining lake and watershed characteristics with Landsat TM data for remote estimation of regional lake clarity, Remote Sensing of Environment, 123, 109-115, 2012.*

*Stadelmann T H, Brezonik P L, Kloiber S M.: Seasonal patterns of chlorophyll a and Secchi disk transparency in lakes of East-Central Minnesota: Implications for design of ground- and satellite-based monitoring programs. Lake and Reservoir Management, 17(4): 299–314, 2001.*

In addition, in order to better explain the reason for choosing summer as the target season, we have modified the relevant descriptions in the manuscript. The details are shown as follows:

P3 Line 125 – P4 Line 131: 'The summer months were chosen because the lake water is abundant and deep, and the water clarity is more stable with fewer short-term variables in summer than in other seasons when there are no climatic events (i.e. heavy rainfall, storm events) (Olmanson et al., 2008; Chen et al., 2021; Song et al., 2020; Singh and Singh, 2015). Additionally, summer is also the period with the most abundance public in-situ water quality

data (Mccullough et al., 2012). Therefore, it is suitable for mapping with remote sensing imagery (Kloiber et al., 2002; Singh and Singh, 2015; Song et al., 2020).'

**Point 3:** Line 163-164. The collected SD measurements were within seven days of satellite overpasses. I suggest the meteorological conditions should be considered since both heavy rain and strong wind could affect the SD.

**Response:** Great thanks for this comment. As you pointed, heavy rain and strong winds can actually affect the SD of lakes. When collecting the measured SD data for validation, it was an oversight that we forgot to consider the meteorological conditions in which the in-situ SD were measured. Now, according to your comments, we have rechecked the weather conditions of each sample, and removed the data on rainy and windy days. A total of 153 SD samples were removed from the validation process, and the final number of samples used for validation was 1349. We have revised the corresponding descriptions to emphasize that only the SD data on sunny days were used for validation. The details are shown below.

P5 Line 160 –Line 165: 'In order to quantitatively evaluate the performance of the LAWSD30 dataset, a total of 1349 in situ SD measurements of 208 lakes between 1992 and 2019 were collected from the China Lake Scientific Database (http://www.lakesci.csdb.cn), the National Earth System Science Data Center, National Science & Technology Infrastructure of China (http://lake.geodata.cn), and work by Wang et al. (2020) and Liu et al. (2020). The collected in-situ SD data were all measured on sunny days.'

P6 Line 176 –Line 180: '

[Figure]

**Figure 2.** Details of the in situ measured SD datasets. (**a–c**) The geographical distributions of SD samples

used to validate the accuracies of the corresponding SD products; (**d**) the probability density of the collected SD measurements used for each target SD product, used to show the SD range where the collected in situ SDs are mainly concentrated.'

P10 Line 304 –Line 313: 'From Fig. 4a, our LAWSD30 dataset exhibited a significant correlation with all collected in situ SD data, with an $R^2$ of 0.92, a $NRMSE$ of 28.1%, and a $MAPE$ of 0.235. Most of the scatter points were distributed close to the 1:1 line. Specifically, from the validation results in the 2010s, our LAWSD30 showed good performance, with an $R^2$ of 0.92, a $NRMSE$ of 30.8%, and a $MAPE$ of 0.268. In addition, a stable performance was also shown in the results for the 2000s, with $R^2$ reaching 0.84, $NRMSE$ reaching 30.1%, and MAPE reaching 0.219. Furthermore, a good performance was also seen before the 2000s, with an $R^2$ of 0.69, a $NRMSE$ of 15.0%, and a $MAPE$ of 0.126 in the 1990s. The validation results for these different decades demonstrated the stable performance of our LAWSD30 in different periods.'

P11 Line 319 –Line 323: '

[Figure]

**Figure 4.** Scatterplots of the in situ measured SD data and our LAWSD30 dataset. (**a**) An overall scatterplot of our LAWSD30 dataset and all the collected in-situ SD data; (**b–d**) scatterplots of our LAWSD30 dataset and the corresponding in situ SD data in the 1990s (1992, 1995, and 1998), 2000s (2001, 2004, and 2007), and 2010s (2010, 2013, 2016 and 2019), respectively.'

**Point 4:** Figure 4. When SD is less than 2, the covariance relationship between in-situ SD and LAWSD30 is very weak because many data are vertical lines. The accuracy should be analyzed in this situation.

**Response:** Great thanks for this comment. By re-checking the data through careful visual inspection, we found that most of the in situ SD data with a weak relationship to LAWSD30 were measurements located in adjacent areas. For example, as shown in Fig. R1, the SD of the entire lake in LAWSD30 dataset is relatively uniform, and the pixels composed of the lake were all composited from the same date. In addition, there were some SD samples collected in the lake. However, since these samples were

collected on different days, and some of them were sampled very close, adjacent water pixels in LAWSD30 with similar SD values may correspond to SD samples of different values collected on different days (such as the two adjacent samples: sample a and sample b in Fig. R3). As a result, some vertical lines appeared in the scatterplot.

[Figure]

| ID | In-situ SD | LAWSD30 | lon | lat | time |
|----|-----------|---------|-----|-----|------|
| a | 1.3 | 1.1044 | 116.3 | 28.50889 | 2008/7/29 |
| b | 1.5 | 1.1038 | 116.3 | 28.51004 | 2008/8/6 |

Figure R1. An example that causes vertical lines to appear

We have revised the manuscript and added the corresponding descriptions. The details are shown below:

P10 Line 313 –Line 317: 'Although some of the measured SD records had some deviations from LAWSD30 and resulted in vertical lines, it was found that these deviations were mainly due to differences between the dates of the SD measurements and the LAWSD30 dataset. Moreover, for most regions, the deviation between the measured SD and LAWSD30 was small.'

**Point 5:** Line 332. Why Selinco Lake and Hongze Lake were chosen to illustrate the LAWSD30?

**Response:** Great thanks for this comment. The first reason for choosing Lake Selinco and Lake Hongze to illustrate LAWSD30 is that these two lakes are distributed in the west and east of China, respectively, which can be used to illustrate the ability of our LAWSD30 dataset in assessing the SD trends of lakes in different environments. Second, since the water environment of the two lakes have been widely concerned (Liu et al., 2021a; Li et al., 2016; Wang et al., 2020; Cao et al., 2017; Li et al., 2019), it is convenient to compare the conclusions of this study with the previous studies.

We have added more descriptions to illustrate the reason for choosing Lake Selinco and Lake Hongze to describe the LAWSD30 in the manuscript. The details are shown as follows:

*Liu, C., Zhu, L., Li, J., Wang, J., Ju, J., Qiao, B., Ma, Q., and Wang, S.: The increasing water clarity of Tibetan lakes over last 20 years according to MODIS data, Remote Sensing of Environment, 253, 2021.*

*Wang, S., Li, J., Zhang, B., Lee, Z., Spyrakos, E., Feng, L., Liu, C., Zhao, H., Wu, Y., Zhu, L., Jia, L., Wan, W., Zhang, F., Shen, Q., Tyler, A. N., and Zhang, X.: Changes of water clarity in large lakes and reservoirs across China observed from long-term MODIS, Remote Sensing of Environment, 247, 2020.*

*Li, N., Shi, K., Zhang, Y., Gong, Z., Peng, K., Zhang, Y., Zha, Y.: Decline in Transparency of Lake Hongze from Long-Term MODIS Observations: Possible Causes and Potential Significance, Remote Sensing, 11, 2019.*

*Cao, Z., Duan, H., Feng, L., Ma, R., Xue, K.: Climate-and human-induced changes in suspended particulate matter over Lake Hongze on short and long timescales, Remote Sensing of Environment, 192, 98-113, 2017.*

*Li, J., Wang, S., Wu, Y., Zhang, B., Chen, X., Zhang, F., Shen, Q., Peng, D., and Tian, L.: MODIS observations of water color of the largest 10 lakes in China between 2000 and 2012, International Journal of Digital Earth, 9, 788-805, 2016.*

P12 Line 344 –Line 349: 'These two lakes were chosen because they are distributed in western and eastern China, respectively, which can be used to illustrate the ability of our LAWSD30 dataset in assessing the SD trends of lakes in different environments. Additionally, since the water environment of the two lakes have been widely concerned and studied (Liu et al., 2021a; Cao et al., 2017; Xue et al., 2019), it is convenient to compare the conclusions of our study with the previous studies.'

**Point 6:** Figure 10. The figures present the different SD trends of lakes with area<=1 km$^2$ and >1km$^2$, in different region of China. The five regions have different socio-economic, geological and climatic conditions, should the driver factors of SD changes be further explained?

**Response:** Great thanks for this comment. According to your comments, we conducted an experiment to briefly analyze the potential linkage between environmental factors and the lake clarity in China. The details are shown below:

In order to preliminarily explore the potential linkage between environmental factors and the spatial-temporal changes of lake clarity in the five limnetic regions, three categories of environmental factors were collected, including: lake area factors (i.e. lake area (LA) of lakes with areas ≤ 1 km$^2$ and the LA of lakes with areas > 1 km$^2$), meteorological factors (i.e. precipitation (PR), wind speed (WS), and near surface air temperature (AT)), and human activity factors (i.e. local gross domestic production (GDP) and population density (PO)). These factors have been considered as potential drivers of the long-term SD trend of lakes in previous studies (Feng et al. 2019a; Li et al. 2020a; Meng et al. 2015; Wang et al. 2020; Yin et al. 2021).

The collected environmental factors were separately counted in each region. Specifically, the annual summer mean values for the PR, WS, and AT in each limnetic region were calculated. Also, the total LA and the intra-year PO and GDP of each limnetic region was calculated for each target year. The long-term variations in these environmental factors were shown in Fig. R2. The GDP, PO, PR, and LA exhibited increase trends in all five limnetic regions from 1985 to 2020. In addition, WS was characterized by increase trends in MXR, YGR, and NER, but downward trends in EPR and QTR. Moreover, AT has been trending upward in EPR, MXR, QTR, and YGR since 1985, but has decreased in NER. The correlation between these factors and

the variations in lake SD was analyzed and shown in Fig. R3.

Regarding the correlations in the east (including EPR and NER), the human activity factors (PO and GDP) showed strong positive correlations with small lakes (with areas ≤ 1 km²) (P < 0.01). Also, a strong positive correlation between GDP and clarity of large lakes (with areas > 1 km²) can be found in NER. In fact, with the development of GDP and urbanization, local and national governments have put forward a variety of regulations to manage water resource and control pollution (Jiang and Zheng 2011; Wang et al. 2020). Our results appeared to further proved the effectiveness of the environmental regulation and projects implemented by the local and national governments. Similarly, previous works also found PO and GDP were positively correlated with the lake clarity in eastern China (Feng et al. 2019a; Guo et al. 2015; Zhao et al. 2018). In addition, the correlation between large lakes and human activity factors was found to be relatively weak in EPR (the correlation coefficient (r) between SD and PO and GDP was 0.12, and -0.09, respectively), which means the regulation and management for small lakes may be stronger than that for large lakes. Furthermore, the small lakes in EPR also showed significant positive correlations with the increase of AT and LA, indicating the positive impact of the two factors on SD of small lakes in EPR. Nevertheless, since warm AT generally promotes the growth of algae, causing algae blooms and reducing water clarity (Li et al. 2020a; Yin et al. 2021), it seems that SD should not increase with the increase in AT. However, because climate, lake area, and human activities are interconnected in complex ways (Maberly and Elliott 2012; Williamson et al. 2009), and other factors (such as GDP, PO, and LA) are also significantly positively correlated with SD, it is possible to lead a positive correlation between AT and SD. Similarly, Wang et al. (2020) and Feng et al. (2019a) also found that some lakes in EPR were positively correlated with lake SD. Then, for the large lakes in EPR, WS and SA showed significant negative and positive correlations with lake clarity, respectively, implying the negative and positive effects of WS and SA on SD in large lakes in this region. Similar results can also be found in recent studies (Feng et al. 2019a; Xue et al. 2019).

In MXR, PO and AT showed significant negative correlations with SD changes in small lakes, suggesting the negative effects of the two factors on SD variations in small lakes. In contrast, all factors exhibited weak and insignificant correlations with large lakes in EPR, implying that large lakes in MXR are more resistant to environmental changes than small lakes.

As for lakes in QTR, the small lakes had significant negative correlations with PO, GDP, AT, and LA, suggesting the negative effects of the four factors on the long-term SD changes in small lakes from 1985 to 2020. Several studies have reported that climate warming has accelerated the melting of glaciers and permafrost in cold regions over recent decades, leading to an increase in the lake area and water volume in QTR (Liu et al. 2021; Sun et al. 2018; Yang et al. 2017; Zhang et al. 2011). Since the sediments carried by meltwater can reduce the SD in lakes (Mi et al. 2011), this is probably the reason for the negative correlations between SD and AT and LA. In addition, because high AT can also affect the growth of phytoplankton in QTR lakes (Feng et al. 2019b), this may also be one of the reasons for the negative effects of AT. In agreement with our results, recent studies have also found the negative effects of AT and LA on lake SD in QTR (Liu et al. 2021; Wang et al. 2020). In contrast, from the results of large lakes in QTR, the correlations between AT and LA and water clarity were shown to be weak, implying that the impact of climate warming on small lakes is significantly stronger than that of large lakes in QTR. The above results can provide effective support for the protection and

governance of the lake environment in this region.

Lastly, for lakes in YGR, small lakes had significant negative correlations with GDP and WS, while none of a factor showed a significant correlation with large lakes. Since the increase in GDP of the catchment may indirectly increase the input of contaminants and nutrients in water bodies (Zhou et al. 2019), and the recovery capacity of small lakes is generally limited (Helen et al. 2015), GDP and the water clarity of small lakes are therefore likely to be negatively correlated. Recently, Xu et al. (2018) found similar correlations between GDP and SD in lakes in YGR. In addition, because wind speed can accelerate the growth of cyanobacteria, resulting in a decrease in the SD of the lake (Shi et al. 2018; Yin et al. 2021), the negative correlation between WS and SD can be well understood and associated with this.

[Figure]

**Figure R2.** Long-term curves of the human activity factors, meteorological factors, and lake area factors in the five limnetic regions (EPR, MXR, QTR, YGR, and NER) from 1985 to 2020. *Noted:* LA., lake area; PR., precipitation; WS., wind speed; AT., near surface air temperature; GDP., gross domestic production; PO., population density.

| | EPR | | MXR | | QTR | | YGR | | NER | |
|---|---|---|---|---|---|---|---|---|---|---|
| | Area ≤ 1 km² | Area > 1 km² | Area ≤ 1 km² | Area > 1 km² | Area ≤ 1 km² | Area > 1 km² | Area ≤ 1 km² | Area > 1 km² | Area ≤ 1 km² | Area > 1 km² |
| PO | 0.97 *** | 0.12 | -0.83 *** | -0.17 | -0.87 *** | -0.42 | 0.11 | 0.48 | 0.79 ** | 0.57 |
| GDP | 0.88 *** | -0.09 | -0.55 | 0.29 | -0.66 * | -0.01 | -0.60 * | 0.44 | 0.88 *** | 0.93 *** |
| PR | 0.34 | -0.28 | -0.18 | 0.18 | -0.51 | -0.50 | -0.01 | 0.08 | 0.22 | 0.23 |
| AT | 0.77 *** | -0.01 | -0.66 * | -0.11 | -0.67 * | -0.32 | -0.48 | 0.19 | 0.36 | 0.41 |
| WS | -0.40 | -0.68 * | -0.49 | 0.38 | 0.21 | -0.06 | -0.62 * | 0.12 | 0.03 | 0.04 |
| LA | 0.68 * | 0.60 * | -0.52 | 0.19 | -0.71 ** | -0.25 | 0.15 | 0.33 | 0.25 | 0.51 |

**Figure R3.** Pearson correlation coefficients (r) between SD and the six environmental factors of lakes with areas > 1 km² and ≤ 1 km² in the five limnetic regions (EPR, MXR, QTR, YGR, and NER).

However, since the focus of our manuscript is on our LAWSD30 dataset and the monitoring of the long-term SD trends of lakes across China, we have not conducted more specific experiments in analyzing the driver factors of SD changes. Also, considering the length of the article and to keep the article focused, we did not incorporate the above analysis into the manuscript. Instead, we added a section "*5.3 limitations and suggestions for future work*" to the Discussion to illustrate the limitations of our study in analyzing the driver factors of SD changes, and to inform readers that investigations into the drivers of SD change are the main focus of our follow-up work. Also, the work of Yu and Zhai (https://doi.org/10.1038/s41598-020-71312-3) was cited in this section to illustrate the possible driving factors that we will analyze. The details are shown below:

P20 Line 543 – P21 Line 559: '

*5.3 limitations and suggestions for future work*

First, due to the relative long revisit period of Landsat series data and the interference from clouds, our study is not suitable for the evaluation of the intra-annual changes in water clarity. The long-term intra-annual and seasonal SD changes of lakes across China still need to be evaluated. In the future, more efforts will be devoted to the use of more multi-source data to increase the number of valid observations, so as to conduct more analysis on SD at seasonal scale.

Second, because this study is mainly focused on monitoring the long-term SD changes in lakes across China, the comprehensive analysis of the relationship between variation of lake water clarity and environmental factors was not investigated. Since the discharge of agricultural fertilizers, industrial sewage, and livestock excrement within the lake drainage area, as well as the spatial-temporal variations of land use and land cover can affect the clarity of the water bodies (Li et al., 2020a), data on related factors should be collected and analyzed. In addition, the topographic conditions of the lake basin, such as the slope of the lake shoreline and the size of recharge river, and the extreme weather and climate events (i.e. drought and heat event) (Liu et al., 2021a; Yu and Zhai, 2020), may also have important impact on the lake clarity. In future work, we will do more investigations to analyze the above potential driver factors of SD changes.'